

# The Influence of HCl on the Evaporation Rates of $H_2O$ over Water Ice in the Range 188 to 210 K at small Average Concentrations

Christophe Delval[1,2,a] and Michel J. Rossi[1,3]

[1]Laboratory of Air and Soil Pollution Studies (LPAS), ENAC Faculty, Swiss Federal Institute of Technology (EPFL), CH-1015 Lausanne, Switzerland

[2]Atmospheric Particle Research Laboratory (APRL), ENAC Faculty, Swiss Federal Institute of Technology (EPFL), CH-1015 Lausanne, Switzerland

[3]Laboratory of Atmospheric Chemistry (LAC), Paul Scherrer Institute (PSI), CH-5232 Villigen-PSI, Switzerland

[a]Present Address: Patent Examiner - Directorate 1657, Dir. 1.6.5.7, European Patent Office, Patentlaan 3-9, 2288 EE Rijswijk, Netherlands

Corresponding author:  michel.rossi@psi.ch

## ABSTRACT

The evaporation flux $J_{ev}(H_2O)$ of $H_2O$ from HCl-doped typically 1.5 μm or so thick vapor-deposited ice films has been measured in a combined quartz crystal microbalance (QCMB) – residual gas mass spectrometry (MS) experiment. $J_{ev}(H_2O)$ has been found to show complex behaviour and to be a function of the average mole fraction $\chi_{HCl}$ of HCl in the ice film ranging from $6 \times 10^{14}$ to $3 \times 10^{17}$ molecule $cm^{-2}$ $s^{-1}$ at 174 – 210 K for initial values $\chi^0_{HCl}$ ranging from $5 \times 10^{-5}$ to $3 \times 10^{-3}$ at the start of the evaporation. The dose of HCl on ice was in the range of 1 to 40 formal monolayers and the $H_2O$ vapor pressure was independent of $\chi_{HCl}$ within the measured range and equal to that of pure ice down to 80 nm thickness. The dependence of $J_{ev}(H_2O)$ with increasing average $\chi_{HCl}$ was correlated with (a) the evaporation range $r^{b/e}$ parameter, that is the ratio of $J_{ev}(H_2O)$ just before HCl-doping of the pure ice film and $J_{ev}(H_2O)$ after observable HCl desorption towards the end of film evaporation, and (b) the remaining thickness $d_D$ below which $J_{ev}(H_2O)$ decreases to less than 85% of pure ice. The dependence of $J_{ev}(H_2O)$ with increasing average $\chi_{HCl}$ from HCl-doped ice films suggests two limiting data sets, one associated with the occurrence of a two-phase pure ice/crystalline HCl hydrate binary phase (set A), and the other with a single phase amorphous HCl/$H_2O$ binary mixture (set B). The measured values of $J_{ev}(H_2O)$ may lead to significant evaporative life-time extensions of HCl-contaminated ice cloud particles under atmospheric conditions, regardless of whether the structure corresponds to an amorphous or crystalline state of the HCl/$H_2O$ aggregate.

## 1. INTRODUCTION

HCl is among the mineral acids that control the acidity of the atmosphere together with $HNO_3$ and $H_2SO_4$. The production of atmospheric HCl is predominantly taking place in the middle and upper stratosphere where $O_3$ is formed owing to photolysis of halogen containing source gases such as CFC's (chlorofluorocarbons). However, there are no known sources of HCl in the upper troposphere (UT) because scavenging processes of HCl throughout the troposphere are very efficient which leads to HCl background concentrations of less than 0.1 ppb (Graedel and Keene, 1995). The absence of significant sources in the troposphere, the long photolytic lifetime of HCl and the fact that the production region is well separated from the regions of interest, namely the UT and the lower stratosphere (LS) all contribute to the fact that HCl is an excellent tracer for stratospheric ozone in the UT (Marcy et al., 2004). Owing to the frequent occurrence of Cirrus clouds in this atmospheric region it is of obvious interest to study the interaction of HCl with atmospheric ice particles at relevant temperature and pressure conditions (Jensen et al. 2001; Zerefos et al., 2003). The compact correlation between $O_3$ and HCl has been used to monitor stratospheric-tropospheric exchange processes and stratospheric $O_3$ intrusions into the troposphere that are still an active field of investigation (Houghton et al, 2001).

HCl is of importance in the LS as it partakes in heterogeneous reactions on Polar Stratospheric Ice Clouds (PSC's) as well as on background stratospheric $H_2SO_4$ aerosol according to the following reaction taken as an example:

$$HCl(ads) + ClONO_2 \rightarrow Cl_2(g) + HNO_3(ads)$$

These reactions efficiently convert inactive Cl-containing reservoir molecules such as HCl and $ClONO_2$ into active photolyzable Cl-containing compounds in a single reaction. Typical examples of such photolabile reaction products are $Cl_2$, $ClNO_2$ and HOCl that will change the atmospheric composition owing to the high reactivity of the photolysis products such as atomic Cl (Solomon et al., 1986; Tolbert et al., 1987; WMO 2003). It thus follows that HCl is of stratospheric importance and is frequently used as a model compound for heterogeneous reactions on ices that has inspired many laboratory kinetic studies (Leu et al., 1991; Hanson and Ravishankara, 1992; Chu et al., 1993; Flückiger et al., 1998; Hynes et al., 2001; Abbatt, 2003).

HCl forms hydrates of variable stoichiometry when exposed to ice depending on the temperature of deposition and the partial pressure of HCl (Ortega et al., 2004; Graham and Roberts, 1997). X-Ray diffraction has allowed the identification of four crystalline hydrates

containing one (Yoon and Carpenter, 1959), two (Lundgren and Olovson, 1967), three
(Lundgren and Olovson, 1967a) and six (Taesler and Lundgren, 1978) $H_2O$ per HCl molecule.
In addition, amorphous mono-, tetra- and hexahydrates have been reported under various
experimental conditions (Yoon and Carpenter, 1959; Delzeit et al., 1993). The control of
growth conditions of a specific HCl hydrate is sometimes elusive, but the formation of a
saturated HCl hexahydrate phase has been reported at sufficiently large HCl exposure
(Graham and Roberts, 1995) using amorphous ice as a starting point despite the fact that the
hexahydrate is said to nucleate with difficulty, at least in thin films (Ortega et al., 2004).
However, the molecular and dynamic details of the crystallization process have not been
investigated as yet.
Fourier-Transform IR (FTIR) absorption measurements have enabled the
characterization of both amorphous as well as crystalline HCl hydrates at growth conditions
that are sometimes significantly different compared to the samples investigated using X-Ray
diffraction. Vibrational spectra of HCl hydrates in the mid IR have been routinely used for
identification purposes for some time (Ferriso and Hornig, 1955; Gilbert and Sheppard,
1973). Recently, the mid IR absorption spectra of the four HCl hydrates mentioned above
have been assigned in a comprehensive and definitive way, albeit without simultaneous proof
of the crystalline structure using X-Ray diffraction (Buch et al., 2002; Xueref and Dominé,
2003). More recently, the reflection absorption IR spectrum (RAIR) of crystalline HCl
hexahydrate in the mid-IR range has been recorded and assigned using theoretical calculations
based on density functional theory that results in a refinement of the geometric structure of
the HCl hydrates and a prediction of the vibrational modes of the crystal (Ortega et al., 2004).
It must be recalled that FTIR spectra in transmission and reflection may in most cases not be
directly compared across the mid IR range.
Regarding the nature of the HCl-ice adsorbate one of the important questions is
whether adsorbed HCl is ionized or exists as a molecular adsorbate under atmospherically
relevant conditions of the UT/LS. This will determine the mechanism of the heterogeneous
reaction which constitutes necessary knowledge for the extrapolation of heterogeneous
reaction rates measured in the laboratory to atmospheric conditions. Thermal desorption of
HCl monitored by IR absorption in the mid-IR range revealed a molecularly adsorbed state of
HCl desorbing below 50 K (Delzeit et al., 1993a). IR studies performed by Banham et al. on
HCl-ice films failed to detect molecularly adsorbed HCl at $T \geq 90$ K despite the high rate of
HCl adsorption in that temperature range (Banham et al., 1995). In contrast, Graham and
Roberts attributed a characteristic Temperature Programmed Desorption (TPD) peak of a
HCl/amorphous ice adsorbate monitored by residual gas MS and occurring at 150 K to
molecularly adsorbed HCl (Graham and Roberts, 1995). However, they did not report the IR
absorption spectrum of the adsorbate in the mid-IR nor did they explain why molecular
adsorption of HCl exclusively occurred on amorphous, but not on crystalline ice. Most recent
results seem to point towards the existence of molecularly adsorbed HCl on ice below 50 K
and at submonolayer coverages in coexistence with ionized solvated HCl whose fraction
increases with increasing ice temperature (Buch et al., 2002; Delzeit et al, 1993a; Delzeit et al.
1997; Uras et al., 1998; Devlin et al., 2002; Lu and Sanche, 2001). Kang et al. discovered
that both molecularly adsorbed as well as ionized HCl coexisted on ice that was deposited
under Ultra-High Vacuum (UHV) conditions in the temperature range 50 to 140 K and under
conditions of low HCl exposure (Kang et al., 2000).
Although theoretical electronic structure calculations predict spontaneous ionization of
adsorbed HCl (Gertner and Hynes, 1996; Bolton and Petterson, 2001) most experiments point
towards a seemingly thermally activated ionization process that may be enabled by structural
factors of the ice matrix that are themselves a function of temperature. Consistent with these
results concentration profiling experiments of HCl/ice adsorbates using static Secondary
Ionization Mass Spectroscopy (SIMS) techniques failed to discover molecularly adsorbed
HCl on ice in the range 90-150 K (Donsig and Vickerman, 1997). In conclusion, both
experimental and theoretical studies clearly point to the absence of significant quantities of
molecularly or covalently adsorbed HCl under stratospheric conditions. Instead, HCl is
ionized and solvated by $H_2O$ on the surface of ice films and may occur either as amorphous
HCl/$H_2O$ hydrates of undefined stoichiometry or as crystalline HCl hydrates. However, these
facts do not rule out the presence of small amounts of molecularly adsorbed HCl on ice that
may be intermediates in the complex mechanism of HCl adsorption on ice as evidenced by
the negative temperature dependence of the rate of uptake of HCl on ice (Flückiger et al.,
1998). In fact, such an intermediate has been invoked in the description of HCl adsorption on
ice under atmospheric conditions using a chemical kinetic model based on a multitude of
experimental observables collected upon HCl uptake on ice (Flückiger and Rossi, 2003).
Work by Parent and coworkers uses Near-Edge X-Ray Absorption Spectroscopy
(NEXAFS) of HCl-doped low temperature ice substrates in order to determine the relative
population of ionic and covalently bound HCl and distinguish between bulk and HCl surface
states in the temperature range 20 to 150 K (Bournel et al., 2002; Parent and Laffon, 2005).
The results seem to confirm the consensus on the low-temperature existence of molecularly
adsorbed HCl up to 90 K beyond which an increasing amount of HCl is converted into an
ionic form, such as $H_3O^+Cl^-$ (Eigen cation) or $H_5O_2^+Cl^-$ (Zundel cation) formed by
spontaneous ionization of adsorbed HCl on ice, up to completion at 150 K (). The newest
work by Parent compares NEXAFS with photoemission (UPS, XPS) and FTIR in
transmission of thin $HCl/H_2O$ films (Parent et al., 2011). The results are roughly consistent
but surprising in the sense that these workers find 92% ionically dissolved HCl in/on ice at 50
K in contrast to Kang et al. (2000) and Devlin et al. (2000) under similar exposure (dose) and
temperature conditions. In addition, Parent et al. (2011) perform the NEXAFS experiment on
a (thick) 100 ML "crystalline" $H_2O$ ice substrate deposited at 150 K whereas the
photoemission and FTIR absorption experiments used a 4 ML thin ice slab deposited at 120
K. The question has to be raised whether the two types of used ice films may be responsible
for some of the discrepancies in the results because both the density and the structure of ice
are known to be a strong function of temperature and deposition conditions (Kuhs et al., 2012;
Schriver-Mazzuoli et al., 2000). The most recent work of Parent et al. (2011) sparked an
interesting controversy in the assignment of the FTIR absorption spectrum of thin $HCl/H_2O$
films and led to two comments showcasing the difficulties of intercomparison of nominally
identical experiments (Devlin and Kang, 2012; Parent et al., 2012).
Furthermore, the results indicate that the "dangling bonds" of the ice surface attributed
to isolated OH groups are not the unique site of HCl adsorption, even in the range 20-90 K
(Flückiger and Delval, 2002). The present work suggests that maiden uptake of HCl onto
pure ice weakens and perturbs the crystal structure of the ice matrix in an irreversible way
such that additional sites for HCl adsorption and ionization are created akin to Parent et al.
(2011). Initial HCl uptake on pure ice therefore has a catalytic effect on the following HCl
uptake. This irreversible nature of initial HCl dosing is known for several years and has been
observed some time ago in Knudsen flow reactor studies on the $HCl/H_2O$ system under
steady-state conditions of both HCl and $H_2O$ at temperatures representative of the UT/LS
(Flückiger et al., 1998; Oppliger et al., 1997). The most recent experimental work on
$HCl/H_2O$ at an atmospherically relevant ("warm") temperature (253 K) has examined the HCl
depth profile using XPS spectroscopy and finds molecularly adsorbed (physisorbed) HCl at its
outermost layer and ionic dissociation in deeper layers (Kong et al., 2017). Complementary
X-Ray absorption results also point towards a perturbation of the crystal structure of ice in the
aftermath of HCl adsorption/dissolution into deeper layers of ice.
We have concluded from recent work that HCl doping in quantities of submonolayer
to several monolayers of HCl leads to the decrease of both the evaporative flux $J_{ev}$ (molecule
$cm^{-2}s^{-1}$) or rate $R_{ev}$ (molecule $cm^{-3}s^{-1}$) and the rate of condensation $k_{cond}$ ($s^{-1}$), of $H_2O$ in the
presence of ice without perturbing the equilibrium vapour pressure of $H_2O$, $P_{H_2O}^{eq}$ (Delval et
al., 2003). We have furthermore shown that the way $J_{ev}$ of $H_2O$ decreases with time depends
on the rate of deposition or the integral of deposited HCl, namely $R_{HCl}$ (molecule s$^{-1}$) and $N_{HCl}$
(molecule), respectively. It appears that two observed HCl species on/in ice, namely single
phase amorphous HCl/$H_2O$ mixtures and a binary phase consisting of pure ice and an as yet
unidentified crystalline HCl hydrate, HCl•$xH_2O$, decrease $J_{ev}(H_2O)$ to a different extent as
proposed in Delval et al. (2003). These results have led us to perform systematic experiments
in this work using the Quartz Crystal MicroBalance (QCMB) combined with residual gas
Mass Spectrometry (MS) that we have used successfully in the past (Delval and Rossi, 2004)
in order to investigate the temporal change of $J_{ev}(H_2O)$ with the increasing average mole
fraction of HCl, $\chi_{HCl}$, remaining in the ice. One of the goals of the present work is to
determine the influence of the HCl deposition parameters on the temporal change of $J_{ev}$ and
the mass accommodation coefficient $\alpha$ during evaporation of a HCl-doped ice film and its
consequence on the lifetime of atmospheric ice particles contaminated by HCl. This issue is
key in relation to the importance of heterogeneous vs. homogeneous atmospheric reactions at
midlatitudes as has been pointed out in the past (Solomon et al., 1986; 1997).
**2. EXPERIMENTAL**
The emphasis of the present experiments was placed on the deposition of small
amounts of HCl ranging in doses from 1 to 40 formal monolayers of HCl where a formal
monolayer of adsorbed HCl corresponded to a surface concentration of $2.5 \times 10^{14}$ molecule cm$^{-}$
$^2$ (Table A1) which is a consensus value obtained from several selected experiments. The
apparatus as well as the methods used for calibration and the HCl deposition procedure have
been described in detail elsewhere (Delval and Rossi, 2005). The experimental conditions are
generally identical to the ones presented in Delval and Rossi (2005) and the instrumental
parameters are summarized in Table 1. The only significant difference between the study of
$HNO_3$-doped ice and the present condensed phase investigation of HCl-doped ice lies in the
mode of trace gas admission. HCl was deposited by backfilling the reactor under stirred flow
conditions with the inlet tubing used for trace gas injection oriented towards one side of the
Si-window of the cryostat set at ambient temperature whereas $HNO_3$ was deposited by
directed injection onto ice films supported by the quartz crystal of the QCMB as referenced
above. Evaporation experiments have been performed isothermally on samples in the
temperature range 174-210 K under dynamic pumping conditions, that is at maximum
pumping speed (gate valve open) in order to prevent readsorption of HCl on the ice substrate.
First, an approximately 1.5 μm thick ice film was grown at 190 K on the quartz crystal
of the QCMB by deposition of bidistilled water vapor at a rate of $1 \times 10^{17}$ molecule cm$^{-2}$ s$^{-1}$
under static conditions. The $H_2O$ equilibrium vapor pressure agreed with published values
across the covered temperature range (Marti and Mauersberger, 1993; Mauersberger and
Krankowsky, 2003). Subsequently, the system was set to the desired temperature given in
Table 2 (second column from the left) and a metered amount of HCl was deposited under
stirred flow conditions. The rate of deposition of HCl, $R_{HCl}$, as well as its time integral,
namely the number of HCl molecules deposited on ice, $N_{HCl}$, have been evaluated using the
method described in Delval and Rossi (2005). Typically, $R_{HCl}$ ranges between $8.0 \times 10^{11}$ and
$4.2 \times 10^{13}$ molecule s$^{-1}$ and $N_{HCl}$ between $1.0 \times 10^{14}$ and $5.4 \times 10^{15}$ molecules. The experimental
conditions of HCl-deposition as well as important experimental parameters are reported in
Table 2. Finally, the system was set to dynamic pumping conditions by opening the gate valve
to the turbopump. $J_{ev}(H_2O)$ was measured isothermally using both the QCBM and residual gas
MS. Figure 1 illustrates a typical experimental protocol of the evaporation at 192 K of a HCl-
doped ice film labelled as experiment 11 in Table 2 and performed as a multidiagnostic
experiment where both the gas- as well as the condensed phases are simultaneously
monitored.
At t = 0, the system is set from stirred flow to dynamic pumping that starts the
evaporation experiment. The continuous curve marked with the empty squares symbol in
Figure 1A corresponds to $J^{QCM}_{ev}$, the evaporative flux of $H_2O$ calculated from the raw signal
at the output of the QCMB. The diamond symbol (◊) corresponds to $J^{18}_{ev}$ evaluated from $I^{18}$,
the MS signal amplitude for $H_2O$ monitored at m/e = 18. Int($J^{18}_{ev}$) marked by triangles in
Figure 1A is the time integral of $J^{18}_{ev}$ and corresponds to the total number of $H_2O$ molecules
that have evaporated from the ice film at t. D is the label at time $t_D$ at which $J_{ev}(H_2O)$
decreased from its original value corresponding to pure ice to 85% of its original value at t =
0, and $d_D$ is the remaining thickness of the ice film at $t_D$. $H_b$ and $H_e$ in Figure 1B correspond to
the time when HCl evaporation begins and ceases to be observed, respectively, using gas
phase residual mass spectrometry (x symbols in Figure 1B) and are labeled $t_{Hb}$ and $t_{He}$. The
data have been treated in analogy to $HNO_3$-doped ice through the formalism given in Delval
and Rossi (2005). Akin to $HNO_3$ the mass balance between HCl deposited, $N^{dep}_{HCl}$, and HCl
recovered during ice evaporation, $N^{evap}_{HCl}$, agrees to within less than a factor of 2 under
dynamic pumping conditions. We therefore estimate the average uncertainty (2σ) of the HCl
mole fraction $\chi_{HCl}$ of $\pm$ 18 % from the average discrepancy between $N^{dep}_{HCl}$ and $N^{evap}_{HCl}$ displayed
in Table 2. In the following $N_{HCl}$ will always refer to $N^{dep}_{HCl}$ derived from the measurement of
HCl at deposition because it refers to a directly measured quantity originating from a
measured pressure decrease in a given volume and time interval $\Delta P/\Delta t$. The present
experiments cover the evaporation of a small albeit important fraction of the model ice film
for which the decrease of $J_{ev}(H_2O)$ is significant.

**3. RESULTS**

The experimental data reported in Table 2 on the isothermal change of the evaporative

flux of water, $J_{ev}(H_2O)$, as a function of the average mole fraction of HCl, $\chi_{HCl}$, in the
remaining ice film during the evaporation process under dynamic conditions are presented in
Figure 2. Dynamic pumping conditions ensure the absence of any readsorption of $H_2O$ vapor
during evaporation owing to the low $H_2O$ partial pressures in the reactor. The axes labelled
"b" and "e" correspond to the values of $J_{ev}(H_2O)$ at the end of ice film deposition and after
desorption of most of the adsorbed HCl from the HCl-doped ice film at $t_{He}$, respectively, as
displayed in Figure 1B. The average mole fraction $\chi_{HCl}$ of HCl in the remaining ice film as a
function of time is calculated according to Delval and Rossi (2005). The change in $\chi_{HCl}$ owing
to $H_2O$ evaporation is evaluated between $t = t_D$ and $t = t_{Hb}$ that corresponds to the time interval
when the number of adsorbed HCl molecules is constant as no release of HCl is observable in
the gas phase at m/e = 36 before $t_{Hb}$. Table 2 also displays the initial value of the HCl mole
fraction, $\chi^0_{HCl}$, calculated for the ice film just at the end of HCl deposition and marked by a
colored circle on the experimental trajectory of a color-coded evaporating ice film displayed
in Figure 2. The average mole fraction of HCl in the ice film, $\chi_{HCl}$, increases owing to
evaporation of $H_2O$ from the ice film without loss of HCl such that the elapsed time increases
with $\chi_{HCl}$ in Figure 2.

The beginning of an evaporation experiment after the end of HCl doping (t = 0 in

Figure 1 or $t_0$ in Figure 2) is marked by a colored circle of a given experiment whose
parameters are displayed in Table 2 and Figure 2 (see experiment 8). As pointed out above, at
$t = t_D$ $J_{ev}(H_2O)$ has decreased to an arbitrarily chosen value of 85% of its original value
measured at $t = t_0$ that corresponds to the beginning of the bold color-coded smooth curve of a
given experiment. Figure 2 essentially displays trajectories of evaporation experiments from $t_0$
(colored circle) moving to $t_D$ and finishing at $t_{Hb}$ between the two limiting values for pure ice
(color coded number of a given experiment on axis "b" for "beginning") and the remaining
ice film at the end of measurable HCl desorption $t_{He}$ (color-coded number of experiment on
axis "e" for "Halogen end"). The trajectory of an experiment with values of $\chi_{HCl}$ between $t_0$
(colored circle at $\chi^0_{HCl}$) and $t_D$ (beginning of bold colored line, see experiment 8 in Figure 2)
ending at $t_{Hb}$ (end of bold line, experiment 8) is presented as a bold dashed-dotted and bold
smooth line from $t_0$ to $t_{Hb}$, respectively, in order to emphasize the quantitative portion of the
experiment. Thinner (color-coded) dotted lines connect the end of ice film deposition (colored
circle on axis "b") and HCl-dosing with $t_0$, the beginning of the evaporation experiment and
also describe the post-phase of evaporation starting at $t_{Hb}$ to $t_{He}$ in order to guide the eye of the
reader to imagine a complete evaporation cycle.
Two different data sets of the change of $J_{ev}(H_2O)$ with $\chi_{HCl}$ may be distinguished in
Figure 2. The first kind of data set corresponds to the curves describing $J_{ev}$ for experiments 1,
2, 9 and 11 and is called dataset A. These traces present a slow continuous decrease of
$J_{ev}(H_2O)$ as $\chi_{HCl}$ increases during $H_2O$ evaporation. The second type of dataset shows an
initial plateau of $J_{ev}(H_2O)$ with increasing $\chi_{HCl}$ starting at the value of pure ice evaporation
followed by a sudden decrease of $J_{ev}(H_2O)$ and is found for experiments 3, 4, 7 and 8 which
we call dataset B. Akin to $HNO_3$, we have evaluated the impact of the HCl deposition
protocol on the evaporation range parameter, $r^{b/e}$, which is the ratio between the evaporative
flux of $H_2O$ at the beginning of ice evaporation, $J^b_{ev}(H_2O)$ reported on the left axis "b" in
Figure 2, and $J_{ev}(H_2O)$ close to the end of the desorption of HCl, $J^e_{ev}(H_2O)$, at $t = t_{He}$ (the right
axis "e" in Figure 2). It describes the factor by which $J_{ev}(H_2O)$ decreases within the limits of
"b" and "e". The impact of both the rate of deposition of HCl on ice, $R_{HCl}$, and its time
integral corresponding to the dose of deposited HCl, $N_{HCl}$, are presented in Figures 3 and
Figure A1 (Appendix), respectively.
It appears from these Figures that we have not succeeded to find a simple experimental
parameter that controls $J_{ev}(H_2O)$ either with elapsed time or amount of adsorbed HCl
expressed as the time dependence of $\chi_{HCl}$. Instead, the data may roughly be classified along
the two cases presented above, namely datasets A and B. The distinction between both data
sets seems to be the rate of change (slope) of $J_{ev}(H_2O)$ within a fairly narrow range of $\chi_{HCl}$.
Indeed, the available number of experiments clearly shows two distinct and limiting cases
whereas the search for other controlling parameters such as $R_{HCl}$, $N_{HCl}$ and the temperature of
deposition ($T_{ice}$) for dataset A failed akin to a similar $HNO_3$ study (Delval and Rossi, 2005).
One may take note for instance of the low value of $\chi_{HCl}$ at 210 K for experiment 9
where the conditions of deposition are similar to experiments 1 and 2, yet, its respective
values of $r^{b/e}$ differ significantly from experiment 9 (Figure 3). In contrast, for dataset B the
$r^{b/e}$ values are similar for the whole set and range from 20 to 27.2 staying within a fairly
narrow band. Moreover, they seem to be independent of $R_{HCl}$ and $N_{HCl}$ as for data set A. In
contrast, the $r^{b/e}$ values for dataset A seem widely scattered over the explored parameter space.
We have also investigated the impact of the deposition protocol on $d_D$, which is the thickness
of ice that is affected by the presence of HCl, namely the remaining thickness of ice whose
$J_{ev}(H_2O)$ value has decreased to 85% of $J_{ev}(H_2O)$ of pure ice. The results on $d_D$ as a function
of $R_{HCl}$ and $N^{dep}_{HCl}$ are presented in Figures 4 and A2 (Appendix), respectively. Taking the
results of Figures 3, 4, A1 and A2 together we arrive at the following two conclusions:
(1) $T_{ice}$, $R_{HCl}$ and $N^{dep}_{HCl}$ are not controlling parameters or predictors for $J_{ev}(H_2O)$ of either set.
(2) The evaporation range parameters $r^{b/e}$ and $d_D$ are not characterizing set A. In contrast, for
dataset B, $r^{b/e}$ and $d_D$ values fall into a narrow range with values varying from 460.7 to 636.0
nm compared to the original ice thickness $d_0$ of 1'500 nm or so (exact numbers in Table 2).

**4. DISCUSSION**

Figure 1 displays the evaporation history of sample 11 as an example whose deposition
parameters are listed in Table 1.  The initial average mole fraction $\chi_{HCl}^0$ of HCl, once
deposition on the 1.44 μm thick ice film under stirred flow reactor conditions is terminated,
has been estimated from the total number of $H_2O$ molecules contained in the ice film and the
measured number of deposited HCl molecules, $N^{dep}_{HCl}$, for experiment 11 (Table 2).  Table 2
and Figure 1 reveal that for approximately $2.2 \times 10^{18}$ $H_2O$ molecules in the film and $5.4 \times 10^{14}$
molecules of deposited HCl, we obtain $\chi_{HCl}^0 = 2.7 \times 10^{-4}$.  This HCl mole fraction represents an
average value that takes into account all $H_2O$ molecules contained in the ice film whereas in
reality there will be a HCl gradient across the ice film as has been observed in the case of the
$HNO_3$/ice system (Delval and Rossi, 2005).
After the HCl deposition process on the typically 1.5 μm thick ice film the gate valve is
opened in order to initiate the isothermal evaporation experiment under dynamic pumping
conditions.  Initially, $H_2O$ evaporates at fluxes $J_{ev}(H_2O)$ that are characteristic of pure ice
measured previously (Delval and Rossi, 2004; Pratte et al., 2006).  These initial values
$J_{ev}^b(H_2O)$ are displayed on the left-hand "b" (= "beginning") axis in Figure 2.  As the
evaporation proceeds $J_{ev}(H_2O)$ slightly decreases with time as displayed in Figure 1A to the
arbitrarily chosen point where $J_{ev}(H_2O)$ has decreased to 85% of the initial pure ice value at
which point the remaining ice thickness $d_D$ has decreased by approximately one third to 771.7
nm remaining ice thickness as displayed in Figure 1B and Table 2.  Further evaporation of
$H_2O$ leads to a continuous decrease of $J_{ev}(H_2O)$ at a corresponding increase of $\chi_{HCl}$ up to point
$H_b$ defined above ("Halogen beginning") at $t_{Hb}$ (Figure 1B) where HCl starts to desorb from
the ice film as monitored using the residual MS signal at m/e = 36.

For $t < t_{Hb}$, $\chi_{HCl}$ is given by the number of originally deposited HCl molecules that

remain adsorbed on the ice film up to $t_{Hb}$ and the remaining $H_2O$ molecules in the film. In
contrast, for $t > t_{Hb}$ the composition of the remaining ice film must be determined by taking
into account the loss by evaporation of both $H_2O$ and HCl. The present experimental
configuration is not adapted to quantitatively measure HCl loss. Therefore, we have chosen to
display the temporal development of $J_{ev}(H_2O)$ for $t < t_{Hb}$ in Figure 2 as a function of the
average value of the HCl mole fraction $\chi_{HCl}$. However, the value of $J_{ev}(H_2O)$ at $t = t_{He}$ where
most of the HCl has desorbed from the ice film is plotted on the right axis labelled "e" (=
"end") as $J_{ev}^{e}(H_2O)$ in Figure 2 in order to provide a limit for the minimum value of the
evaporation rate $J_{ev}(H_2O)$ at an ice film thickness $d_{He}$ of approximately $80 \pm 10$ nm as
displayed in Figure 1B. We have observed in the past that $J_{ev}(H_2O)$ for a pure ice film of
approximate thickness of 80 nm or less also slows down, presumably owing to island
formation at the very end of pure thin ice film evaporation (Delval and Rossi, 2005).
Therefore, results are becoming more difficult to interpret such that we halted the experiment
at $t_{He}$. The ratio $r^{b/e} = J_{ev}^{b}(H_2O)/J_{ev}^{e}(H_2O)$ is displayed in Table 2 and is an operational
evaporation range parameter that estimates the extent of decrease of $J_{ev}(H_2O)$ for a thick HCl-
doped ice film of μm size down to thicknesses of approximately 80 nm.

At the start of the evaporation experiment the equilibrium vapor pressure of $H_2O$,

$P_{eq}(H_2O)$, is that of pure ice (Delval et al., 2003; Delval and Rossi, 2004; Pratte et al., 2006)
owing to the small values of $\chi_{HCl}^{0}$. Raoult's Law applies to such small values of $\chi_{HCl}$ but leads
to unmeasurably small deviations from the observed vapor pressure of $H_2O$ which is that of
pure ice. In fact, we have never observed an equilibrium vapor pressure that did not
correspond to pure ice in the course of the present work that seems to be the consequence of
the small average mole fractions of HCl in the $H_2O$/HCl system. This value of $P_{eq}(H_2O)$ is
observed throughout the evaporation up to $t_{He}$ as the film is apparently sufficiently $H_2O$-rich
to support an equilibrium vapour pressure characteristic of pure ice consistent with the
published, albeit revised HCl/$H_2O$ phase diagram by Iannarelli and Rossi (2014).  In view of
the decreasing values of $J_{ev}(H_2O)$ displayed in Figure 2 the equilibrium vapour pressure of
pure ice can only be maintained if the condensation rate coefficient $k_c$ for $H_2O$ adsorption
decreases to the same extent as $J_{ev}(H_2O)$ in agreement with previous work (Delval et al.,
2003; Delval and Rossi, 2004; Pratte et al., 2006) and the concept of microscopic
reversibility.
Figure 1A displays both the QCMB signal ($\square$) as well as the corresponding MS signal
for evaporating $H_2O$ at m/e = 18 ($\lozenge$).  Akin to the $HNO_3/H_2O$ system studied previously
(Delval and Rossi, 2005) we obtain a perfect match between the two signals for $t < t_D$ whereas
for $t > t_D$ there is a significant discrepancy, especially at t > 300 s amounting to typically less
than a factor of two.  Such a disagreement has been noted before for $HNO_3/H_2O$, albeit to a
larger extent.  The reason for this behaviour of the QCMB signal has not been studied in
detail but may well lie in a structural rearrangement of the condensed phase during
evaporation that will lead to a change in the calibration factor $C_f$ defined in Table 1 and in
Delval and Rossi (2005).  In view of the straightforward interpretation of the calibrated MS
signal at m/e = 18 we have used it for the measurement of $J_{ev}(H_2O)$ at $t > t_D$ akin to the
previous study on $HNO_3/H_2O$.
The accuracy with which both $t_{Hb}$ and $t_{He}$ can be determined depends on the temporal
change of the background MS signal for HCl at m/e = 36 displayed in Figure 1B following
the dosing of the thin ice film under stirred flow conditions.  Figure 1B displays the MS signal
at m/e = 36 as a function of time just before the start of HCl desorption at $t_{Hb}$ that is signalled
by an increase in the MS intensity whereas $t_{He}$ corresponds to the return of the HCl signal to
the decaying HCl background in comparison to a reference experiment in which the HCl
background was monitored as a function of time following the admission of the same HCl
dose in the absence of an ice film.  We estimate that $t_{Hb}$ is determined to $\pm$ 10 s whereas $t_{He}$
may only be estimated to $\pm$ 100 s by virtue of the vanishing intensity of the HCl MS signal
compared to its slowly decaying background.
Previous work has established that the rate of deposition of HCl, $R_{HCl}$, in the range
$1 \times 10^{13}$ to $5 \times 10^{13}$ molecule $s^{-1}$ for the 0.78 $cm^2$ surface area of the Si-window leads to the
formation of a crystalline HCl hydrate, $HCl \cdot xH_2O$, whereas values outside of this range
seemed to favor the formation of an amorphous $HCl/H_2O$ mixture (Delval et al., 2003).  The
exact nature of this undoubtedly crystalline solid is still unknown. However, IR spectroscopic
work on hydroxonium salts of the type $H_3O^+X^-$ suggests that the $\nu_1$ and $\nu_3$ peak positions of
the symmetric and antisymmetric O-H stretch vibrations must correspond to a molecular
structure in which the distance between the cation and anion is unusually large (Desbat and
Huong, 1975; Iannarelli and Rossi, 2016). Recent work has shown that the presence of HCl
hexahydrate ($HCl \cdot 6H_2O$) under the present experimental conditions could be safely excluded,
however, the FTIR absorption spectrum clearly shows the presence of dissociated HCl within
the ice film (Iannarelli and Rossi, 2014). Akin to HCl•6H$_2$O that is known to nucleate with
difficulty, crystallization of this unknown HCl hydrate seems to occur only under specific
conditions of temperature and/or HCl deposition. Owing to the quantitative control of HCl
deposition on the ice film in this work we infer the presence of at least two forms of HCl
hydrates in the temperature range chosen in analogy to previous work (Delval et al., 2003).

We clearly point out that the present work has been performed without simultaneous

spectroscopic control of the HCl/ice deposit that would have allowed the identification and/or
quantification of the molecular composition of the condensate. Because we lack a
spectroscopic probe for the ice film deposited on the QCMB in the present work we are
seeking a correlation between the type of HCl/H$_2$O deposit, either crystalline or amorphous,
and the relevant HCl deposition parameters. Previous work has revealed a distinctly different
temporal dependence of $J_{ev}(H_2O)$ between the crystalline and amorphous HCl hydrates with
the extent of H$_2$O evaporation from the film, both at low (Delval et al., 2003) and high
temporal resolution (Iannarelli and Rossi, 2014).

Datasets A and B have been characterized above in terms of a difference in the temporal

dependence of $J_{ev}(H_2O)$ as a function of increasing $\chi_{HCl}$ owing to H$_2$O evaporation. Taking
one example of each set Figure 2 reveals a distinct difference between experiment 7 (set B)
and 11 (set A) performed at T = 195 and 192 K, respectively, despite comparable HCl
deposition parameters (Table 2). At t > t$_D$ $J_{ev}(H_2O)$ for experiment 7 decreases at once with
$\chi_{HCl}$ in contrast to experiment 11 whose $J_{ev}(H_2O)$ value gradually starts to decrease at roughly
the same value of $\chi_{HCl}$ as experiment 7. In addition, in both cases the extent of the decrease of
$J_{ev}(H_2O)$ is roughly equal between t$_D$ and t$_{Hb}$ within less than a factor of two. Set B data are in
marked contrast to set A independent of the magnitude of $\chi_{HCl}$ which is highlighted by a
comparison of experiment 11 (set A) and 4 (set B) at 192 and 190 K, respectively. The abrupt
decrease of $J_{ev}(H_2O)$ for set B as well as the gradual decline for set A both at t$_D$ occur before
HCl starts to evaporate from the sample at t$_{Hb}$ and appear therefore to be independent of $\chi_{HCl}$
within the range explored in the present work.

If we consider the mean value <d$_D$> for data set B (Figures 4 and A2) we find 549.0 ±

120.0 nm compared to the 1'500 nm or so original ice thickness which corresponds to
approximately 8.5x10$^{17}$ molecules of H$_2$O spread out over 0.50 cm$^2$. These H$_2$O molecules
are impacted by the presence of HCl to some extent because $J_{ev}(H_2O)$ is slowed down
significantly compared to pure ice. Previous results (Delval et al., 2003) on the deposition of
HCl on ice under conditions where the presence of an as yet unidentified crystalline hydrate
HCl•xH$_2$O was confirmed by FTIR absorption led to the conclusion that on average the
amount of "trapped" $H_2O$ within $d_D$ corresponded to $1.2x10^{18}$ molecules starting with an
original 1 μm thick ice film that was subsequently doped with HCl.  This quantity of $H_2O$,
when scaled from the 0.78 $cm^2$ area of the Si-window used for FTIR absorption to the area of
0.5 $cm^2$ of the QCMB leads to $7.7x10^{17}$ $H_2O$ that is in satisfactory agreement with the present
measurement of $d_D$ or $8.5x10^{17}$ $H_2O$ in the present work.  We may add that the previous value
of $1.2x10^{18}$ $H_2O$ from the work of Delval et al. (2003) corresponding to $d_D$ obtained in that
work has been derived using HeNe interferometry which is a crude method for measuring the
film thickness.
Specifically, considering the low value of $d_D$ of experiments 1 and 10 (Table 2, Figure
4) we may define the behaviour of these condensates as "ice-like" because roughly 80% of the
ice sample of roughly 1.5 μm thickness has evaporated at $J_{ev}(H_2O)$ of pure $H_2O$ ice before it
slows down. This decrease of $J_{ev}(H_2O)$ is a kinetic effect and acts on both the rate of
evaporation as well as on the mass accommodation coefficient, the ratio of which remains
constant because the characteristic vapor pressure of pure ice is maintained until $t = t_{He}$ when
the sample runs out of $H_2O$ and HCl. For sample 1 this conclusion is not too surprising owing
to its extremely low HCl dose of 0.8 formal HCl monolayers. Sample 10 in comparison with
the other members of data set A allows us to conclude that $d_D$ is proportional to $T_{ice}$ for data
set A. Low temperatures prevent rapid diffusion of HCl into the bulk of the ice film which
leaves the majority of the total mass of the thin film deposited void of any HCl. Therefore, a
large fraction of the total mass of the thin film deposit evaporates at values of $J_{ev}(H_2O)$
characteristic of pure ice before it decreases to lower values when the presence of HCl slows
down $J_{ev}(H_2O)$. Although our experiment does not reveal the location of the thin layer of HCl-
contaminated ice, plausibility suggests that it is located on top of the ice film at the gas-
condensed interface. The corollary of this is that it is impossible to "cap" a pure ice sample
with a thin layer of an atmospheric condensable gas of lower vapor pressure in the hope to
lower the vapour pressure of the condensate or slow down $H_2O$ evaporation. This capping has
been attempted many times, and examples abound. However, all attempts to lower the ice
vapor pressure of the condensate using low amounts of polar contaminants of ice, such as
$HNO_3$, HCl or HBr have proven futile to date (Biermann et al., 1998).
The other members of data set A are examples (experiments 2, 9, 11) with high values
of $d_D$ at higher temperatures and higher HCl doses (Table 2). Because of higher presumed
interfacial HCl concentrations these samples experience a decrease in $J_{ev}(H_2O)$ owing to rapid
diffusion of HCl into ice that affects the kinetics of evaporation to some depths of the ice film
corresponding to higher values of $d_D$. Both high HCl doses and high temperatures favor HCl
contamination of deeper layers of the HCl film, hence high values of $d_D$.
Tentatively, we assign a crystalline, yet unknown molecular structure and stoichiometry
to samples A in contrast to samples of dataset B that we identify with an amorphous structure
in terms of a liquid HCl/$H_2O$ mixture of variable composition. The main argument in favour
of this assignment comes from recent kinetic work performed by Iannarelli and Rossi (2016a)
who show that both $J_{ev}(H_2O)$ as well as the corresponding mass accommodation coefficient or
the adsorption rate coefficient for $H_2O$ adsorption is highly scattered for crystalline HCl
hexahydrate whereas the amorphous mixture shows a significantly smaller scatter of the
experimental and thermodynamic values (Iannarelli and Rossi, 2014). Figure A3 and A4 in
the Appendix show this substantial difference in experimental scatter for the amorphous
HCl/$H_2O$ mixture (Figure A3) compared to crystalline HCl hexahydrate (Figure A4).
Figure 3 displays the range parameter $r^{b/e}$ as a function of $R_{HCl}$ for all data displayed in
Table 2. It is noteworthy that $r^{b/e}$ is in the range 20 to 27 for set B experiments 3, 4, 7 and 8
compared to set A data that seem to be scattered throughout the range. Members of data set B
show a common average range for both $d_D$ and $r^{b/e}$ which is the reason we tentatively assign
these structures to amorphous liquid mixtures of high viscosity at the prevailing temperatures.
In conclusion, we take the simultaneous occurrence of the restricted range of the
measured remaining thickness of ice $d_D = 549.0 \pm 120.0$ nm together with a similarly
restricted range of $r^{b/e}$ between 20 and 27 as well as the substantial overlap in $R_{HCl}$ between
the present and previous work (Delval et al., 2003) as an indication that set B evaporation
experiments imply the presence of an amorphous HCl/$H_2O$ mixture. In contrast, the scatter of
the set A data across the range of $r^{b/e}$ and $d_D$ values suggests the presence of an as yet
unidentified crystalline HCl hydrate. If, and only if the HCl deposition conditions rapidly
establish thermodynamic equilibrium, experiment 2 (low HCl flow rate) lies in the "ice"
region in the temperature interval 192-210 K whereas experiment 11 (high HCl flow rate)
should access crystalline HCl hexahydrate at 192 K but not at 210 K according to the revised
HCl/$H_2O$ phase diagram of Iannarelli and Rossi (2014). It remains to be seen whether or not
the published FTIR absorption spectrum in Delval et al (2003) turns out to be identical to the
expected crystalline HCl hexahydrate invoked as condensate in set A molecules, similar HCl
deposition parameters notwithstanding. This proposal awaits further confirmation from FTIR
spectroscopic work that will be combined in the future with the QCMB measurement. At this
point we reiterate our earlier statement that $T_{ice}$, $R_{HCl}$, $N^{dep}_{HCl}$ do apparently not control
$J_{ev}(H_2O)$ of both datasets..

## 5. ATMOSPHERIC IMPLICATIONS

The evaporation range parameter $r^{b/e}$ may be used to quantitatively evaluate the upper limit of the evaporative lifetime extension of thin ice films under conditions of $H_2O$ vapor subsaturation. In the interest of applying the data of the present work to atmospheric conditions we make the assumption that typical atmospheric Cirrus cloud particles of several µm diameter may be approximated by macroscopic thin films used to obtain the present data. The time $t_{ev}$ in seconds to complete evaporation of an ice particle of radius r at a given relative humidity (rh) is given in equation 1 (Chiesa and Rossi, 2013; Iannarelli and Rossi, 2016a):

$$t_{ev} = \frac{(\frac{\rho N_L}{M})^{2/3}(\frac{r}{a})}{J_{ev}(1-rh)} \tag{1}$$

where $\rho$ is the density of ice (0.916 and 0.925 g cm$^{-3}$ at 273 and 173 K, respectively), M = 18 g mol$^{-1}$ for $H_2O$, r and a are the ice particle radius and the distance between two molecular layers in $H_2O$(ice), respectively (Iannarelli and Rossi, 2016a). Equation 1 is based on a simple layer-by-layer evaporation model of $H_2O$(ice) from a spherical ice particle following a zero-order rate law for $J_{ev}$ or a first order rate law for its inverse, namely $H_2O$ adsorption or condensation. For a 10 µm diameter ice particle approximated by thin film experiment 1 (Table 2) at rh = 80%, T = 192 K, $J_{ev} = 3\times10^{16}$ molecule s$^{-1}$ cm$^{-2}$ (Petrenko and Whitworth, 1999) and a = $4\times10^{-8}$ cm we obtain $t_{ev}$ = 2050 s or 34 min. This is the value for a pure ice particle as $J_{ev}(H_2O)$ for pure ice has been used at the outset of the evaporation experiment and is a lower limit to the true evaporation time owing to the competition of mass transfer and heterogeneous chemistry (Seinfeld and Pandis, 1998). Using $r^{b/e}$ = 43 for experiment 1 $t_{ev}$ is calculated to be 15 minutes and 24 hours for a 100 nm and 10 µm diameter particle, respectively, whereas the evaporative lifetime of an analogous pure ice particle would be only 21 s for the 100 nm diameter pure ice particle. Cirrus ice particles are frequently in the lower tens of µm size range resulting in a longer evaporation time considering that the simple evaporation model scales linearly with the radius of the ice particle. In conclusion we may state that, owing to the lifetime extension of ice particles contaminated by HCl, $HNO_3$ or other volatile atmospheric trace gases such as HOCl, HOBr or HONO, small particles may have a chance to survive subsaturated regions of the atmosphere so as to function as cloud condensation or ice nuclei for the following cloud cycle (Delval and Rossi, 2004; 2005; Pratte et al., 2006).

We would like to stress, that the variable $r^{b/e}$ factor displayed in Table 2 leads to a significant increase in the evaporative lifetime of a contaminated ice particle and amounts to a **kinetic effect** that does not affect the equilibrium vapor pressure of the ice particle in question: it is that of pure ice from the start of the evaporation experiment to $t = t_{He}$ and therefore affects both the rate of evaporation and accommodation equally. However, in cases the sample has lost most of its mass the vapor pressure decreases and becomes somewhat uncertain. In the present case the above statement is correct for $t = t_{Hb}$, that is before halogen evaporation. Of note is the fact, that the accommodation coefficient $\alpha$ is frequently less than unity, in contrast to what is often assumed, which will lower the rate of evaporation for pure ice, hence increase the evaporative lifetime of pure ice particles for $T \geq 180$ K as proposed in previous work (Delval and Rossi, 2004; 2005; Pratte et al., 2006).

As a token example of potential atmospheric importance of the measured evaporative lifetimes of ice particles laced with condensable atmospheric trace gases we may take the formation, persistence and evaporation of contrails and Cirrus clouds in the UT/LS. These are ice clouds forming on non-volatile ice nuclei at the corresponding temperature and relative humidity conditions and also frequently serve as reaction sites for heterogeneous atmospheric reactions in connection with ozone depletion and chlorine activation chemistry in the LS. Under certain conditions, Schumann and coworkers used the concept of the increase of the evaporative lifetimes of contaminated ice particles in aviation contrails occurring mostly in the UT, but sometimes also in the LS, in order to explain the persistence of ice clouds below ice saturation conditions up to a certain time duration. Ice clouds have a significant radiative forcing effect that is of interest in evaluating the climate forcing of high-flying aircraft in future aviation scenarios (Lewellen, 2014; Schumann et al, 2017; 2017a). However, the results of the present work show that the rate of evaporation of ice films doped with small amounts of acidic trace gases significantly slows down in a complex manner over the evaporation history of the film or particle, and that the application of equation (1) to atmospheric situations should be carried out with caution.

**CONCLUSIONS**

Despite the scatter of the values of $r^{b/e}$ and $d_D$ in dataset A displayed in Figures 3 and 4 and the apparent lack of influence of the deposition parameters ($T_{ice}$, $R_{HCl}$, $N^{dep}_{HCl}$) on $J_{ev}(H_2O)$ we may state several key points from the present work:

(a) We observe two types of behaviour, both complex, as far as the temporal change of $J_{ev}(H_2O)$ with on-going evaporation of $H_2O$ from a $HCl/H_2O$ condensate is concerned. We have named it sets A and B that represent limiting behaviour as not all performed experiments fit into this scheme.

(b) At low temperature or low dose of deposited HCl ($N^{dep}_{HCl}$) set A samples, especially samples 1 and 10, reveal an "ice-like" behaviour that corresponds to a low value of $d_D$. This means that the $HCl/H_2O$ condensate evaporates a large fraction of the sample thickness at a value of $J_{ev}(H_2O)$ characteristic of pure ice before slowing down at increasing mole fraction of HCl upon $H_2O$ evaporation. This corresponds to a two-phase system consisting of a major ice-like and a minor $HCl/H_2O$ phase having both significantly different values of $J_{ev}(H_2O)$.

(c) High values of $d_D$ are observed at high $T_{ice}$ or $N^{dep}_{HCl}$ values for set A samples. This means that the sample evaporates $H_2O$ at $J_{ev}(H_2O)$ characteristic of pure ice for a relatively short time of its evaporation history because the quantity of HCl is sufficient to decrease $J_{ev}(H_2O)$ already at high values of $d_D$ by rapidly diffusing to deeper layers of the ice film. An equivalent way of expressing the point would be to state that $d_D$ which is an indicator of the total mass of the ice film, is proportional to $T_{ice}$ for Set A.

(d) Set A samples generally show scattered values of both $d_D$ and $r^{b/e}$ values that we attribute to the existence of a two-phase binary system, namely a pure ice and a crystalline HCl hydrate phase of as yet unknown stoichiometry $HCl \bullet xH_2O$, but probably HCl Hexahydrate. At first the pure ice phase starts to evaporate as a whole for a fairly long time at characteristic values of $J_{ev}(H_2O)$ until the pure ice phase has disappeared, followed by the crystalline $HCl/H_2O$ phase at a lower rate of $J_{ev}(H_2O)$ to attain the characteristic value for the evaporation of the crystalline $HCl \bullet xH_2O$ phase.

(e) Set B samples are tentatively identified as single phase binary amorphous mixtures of $HCl/H_2O$ whose kinetic properties are uniform, thus fairly independent of the HCl concentration at the gas-condensed phase interface. The observation of a medium size average value for both $r^{b/e}$ and $d_D$ is consistent with these observations and manifests itself as a continuous, yet gradual decrease of $J_{ev}(H_2O)$ with increasing $\chi_{HCl}$. It is in distinct contrast to Set A where $J_{ev}(H_2O)$ values are those of pure ice until the ice phase has completely evaporated followed by a gradual decline of $J_{ev}(H_2O)$ when the crystalline HCl hydrate starts to decompose.

(f) It must be recalled that the vapour pressure of $H_2O$ remained that of pure ice during
most of the thickness of the $H_2O$/HCl condensate down to approximately 80 nm at
which point we halted the evaporation experiment. This result is expected based on
Raoult's law owing to the small average HCl mole fractions in doped ice used in the
present work: It would make the decrease of the $H_2O$ saturation vapour pressure
unmeasurably small. The present results therefore primarily address the **kinetics** of
$H_2O$ evaporation which changes with the total mass of the thin film condensate and
the concomitant increase in HCl concentration and/or mole fraction.

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

1644.



The authors declare no competing interests regarding the present work.

**Acknowledgement**
We sincerely thank Dr. R. Iannarelli for Figures A3 and A4 displayed in the Appendix. We
also would like to thank the Swiss National Science Foundation (SNSF) for unfailing support
over the years. This work has been performed under SNSF grants no. 20-65299.01 and
200020-105471.


| Table 1: Hardware parameters of both cryogenic sample supports | Si Optical Window | QCM |
|---|---|---|
| Reactor temperature $T_r$ [K] | 320 | |
| Reactor volume $V_r$ [cm$^{-3}$] | 2350 | |
| Conversion factor (1/RT) Conv [molec cm$^{-3}$ Torr$^{-1}$] with R=62398 [Torr cm$^3$ mol$^{-1}$ K$^{-1}$] | $3.0 \cdot 10^{16\,(1)}$ | |
| Sample surface area [cm$^2$] | 0.78 | 0.50 |
| $H_2O$ collision frequency with ice sample $\omega_{H2O}$ [s$^{-1}$] | 5.08 | 3.26 |
| $H_2O$ effusion rate constant of calibrated leak $k_{esc}(H_2O)$ [s$^{-1}$] | 0.064 | |
| MS calibration factor for $H_2O$ (m/z=18, Stirred Flow) $C_{18}^{S-Flow}$ [molec s$^{-1}$ A$^{-1}$] | $2.4 \cdot 10^{24}$ | |
| MS calibration factor for $H_2O$ (m/z=18, Dynamic ) $C_{18}^{dyn}$ [molec s$^{-1}$ A$^{-1}$] | $1.7 \cdot 10^{25}$ | |
| HCl collision frequency with ice sample $\omega_{HCl}$ [s$^{-1}$] | 3.59 | 2.31 |
| HCl effusion rate constant of calibrated leak $k_{esc}(HCl)$ [s$^{-1}$] | 0.047 | |
| MS calibration factor for HC1 (m/z=36, Stirred Flow) $C_{36}^{S-Flow}$ [molec s$^{-1}$ A$^{-1}$] | $3.9 \cdot 10^{24}$ | |
| MS calibration factor for HC1 (m/z=36, Dynamic) $C_{36}^{dyn}$ [molec s$^{-1}$ A$^{-1}$] | $6.3 \cdot 10^{24}$ | |
| Calculated escape orifice area $A_{esc}$ [mm$^2$] | 1.0 | |
| | $d = 10^4$ Å or 1.0 μm for O.D.= 1.08 [A][2] at 3260 cm$^{-1}$ | Calibration Factor |

| Temperature [K] | ratio[3] |
|---|---|
| 170 | 9.0 |
| 180 | 8.0 |
| **190** | **7.8** |
| 193 | 6.0 |
| 205 | 2.0 |
| 208 | 1.9 |

[1] Wall temperature of the reactor at T = 320 K

[2] See (Delval et al., 2003)

[3] Corresponds to the ratio between the true number of molecules present on the QCM support and the number of molecules displayed by the IC5 controller (Delval et al., 2004)

**Table 2 : Representative experimental results for the kinetics of H₂O evaporation in the presence of HCl for increasing HCl deposition temperatures at given rates of deposition $R_{HCl}$ and doses of HCl $N_{HCl}^{dep}$. In the first column the number refers to the corresponding experiment and identifies the data displayed in Figure 2**

| Experiment number | $T_{ice}$ | $d_0$ | $N_{H2O}^0$ | $R_{HCl}$ | $t_{dep}$ | $N_{HCl}^{dep}$ | HCl | $N_{HCl}^{evap}$ | $\chi_{HCl}^0$ | $d_D$ | $J_{ev}^b$ | $J_{ev}^e$ | $r^{b/e}$ |
| --- | --- | --- | --- | --- | --- | --- | --- | --- | --- | --- | --- | --- | --- |
| | [K] | [Å] | [molec] | [molec s$^{-1}$] | [s] | [molec] | ML | [molec] | | [Å] | [molec cm$^{-2}$ s$^{-1}$] | [molec cm$^{-2}$ s$^{-1}$] | |
| **10** | 174 | 15230 | $2.4 \cdot 10^{18}$ | $6.4 \cdot 10^{12}$ | 94 | $6.0 \cdot 10^{14}$ | 4.8 | $4.7 \cdot 10^{14}$ | $2.5 \cdot 10^{-4}$ | 2733 | $1.9 \cdot 10^{15}$ | $4.4 \cdot 10^{14}$ | 4.3 |
| **5** | 188 | 13318 | $2.0 \cdot 10^{18}$ | $1.3 \cdot 10^{13}$ | 66 | $8.7 \cdot 10^{14}$ | 7.0 | $8.9 \cdot 10^{14}$ | $4.4 \cdot 10^{-4}$ | 4540 | $1.2 \cdot 10^{16}$ | $3.9 \cdot 10^{15}$ | 3.1 |
| **4** | 190 | 14016 | $2.1 \cdot 10^{18}$ | $4.2 \cdot 10^{13}$ | 126 | $5.4 \cdot 10^{15}$ | 43.2 | $3.6 \cdot 10^{15}$ | $2.6 \cdot 10^{-3}$ | 6360 | $2.9 \cdot 10^{16}$ | $1.4 \cdot 10^{15}$ | 20.7 |
| **6** | 190 | 13886 | $2.1 \cdot 10^{18}$ | $3.9 \cdot 10^{13}$ | 56 | $2.2 \cdot 10^{15}$ | 17.6 | $1.8 \cdot 10^{15}$ | $1.0 \cdot 10^{-3}$ | 12861 | $3.4 \cdot 10^{16}$ | $1.7 \cdot 10^{16}$ | 2.0 |
| **1** | 192 | 14926 | $2.3 \cdot 10^{18}$ | $3.1 \cdot 10^{12}$ | 36 | $1.0 \cdot 10^{14}$ | 0.8 | $1.8 \cdot 10^{14}$ | $4.3 \cdot 10^{-5}$ | 2823 | $2.9 \cdot 10^{16}$ | $7.1 \cdot 10^{14}$ | 40.8 |
| **2** | 192 | 14682 | $2.3 \cdot 10^{18}$ | $8.0 \cdot 10^{11}$ | 356 | $2.6 \cdot 10^{14}$ | 2.1 | $1.6 \cdot 10^{14}$ | $1.1 \cdot 10^{-4}$ | 6817 | $3.2 \cdot 10^{16}$ | $6.5 \cdot 10^{14}$ | 49.2 |
| **11** | 192 | 14420 | $2.2 \cdot 10^{18}$ | $5.4 \cdot 10^{12}$ | 108 | $5.4 \cdot 10^{14}$ | 4.3 | $6.8 \cdot 10^{14}$ | $2.4 \cdot 10^{-4}$ | 7717 | $4.0 \cdot 10^{16}$ | $7.9 \cdot 10^{14}$ | 50.6 |
| **3** | 193 | 14423 | $2.2 \cdot 10^{18}$ | $3.5 \cdot 10^{12}$ | 220 | $7.0 \cdot 10^{14}$ | 5.6 | $8.1 \cdot 10^{14}$ | $3.2 \cdot 10^{-4}$ | 5659 | $4.9 \cdot 10^{16}$ | $1.8 \cdot 10^{15}$ | 27.2 |
| **7** | 195 | 12614 | $1.9 \cdot 10^{18}$ | $4.3 \cdot 10^{12}$ | 45 | $1.9 \cdot 10^{14}$ | 1.5 | $1.8 \cdot 10^{14}$ | $1.0 \cdot 10^{-4}$ | 5325 | $4.6 \cdot 10^{16}$ | $2.0 \cdot 10^{15}$ | 23.0 |
| **8** | 205 | 13505 | $2.1 \cdot 10^{18}$ | $1.6 \cdot 10^{13}$ | 36 | $5.9 \cdot 10^{14}$ | 4.7 | $3.0 \cdot 10^{14}$ | $2.8 \cdot 10^{-4}$ | 4607 | $2.0 \cdot 10^{17}$ | $1.0 \cdot 10^{16}$ | 20.0 |
| **9** | 210 | 13134 | $2.0 \cdot 10^{18}$ | $3.5 \cdot 10^{12}$ | 84 | $3.0 \cdot 10^{14}$ | 2.4 | $1.9 \cdot 10^{14}$ | $1.5 \cdot 10^{-4}$ | 12136 | $3.0 \cdot 10^{17}$ | $1.8 \cdot 10^{16}$ | 16.7 |

**Figure Captions**

Figure 1: Typical experimental protocol of the evaporation at 192 K of an approximately 1.2 μm thick ice film doped with $5.4 \cdot 10^{14}$ molecules of HCl. This illustration corresponds to experiment 11 of Table 2. (○): ice thickness monitored by QCM (Å), (□): "apparent" $H_2O$ evaporative flux, $J^{QCM}_{ev}$, monitored by QCM (molecule cm$^{-2}$ s$^{-1}$), (+): $I^{18}$ MS signal for $H_2O$, (×): $I^{36}$ MS signal for HCl (A), (◊): $J^{18}_{ev}$ evaporative flux calculated from $I^{18}$ (molecule cm$^{-2}$ s$^{-1}$), (Δ): Int($J^{18}_{ev}$) time integral of $J^{18}_{ev}$ (molecule cm$^{-2}$).

Figure 2: Change of the evaporative flux $J_{ev}(H_2O)$ as a function of the HCl mole fraction ($\chi_{HCl}$) for the cases presented in Table 2 color-coded according to the corresponding experiment number in Table 2. The colored and circled numbers on axis "b" (left) correspond to $J_{ev}(H_2O)$ of pure ice before HCl deposition, the ones on axis "e" (right) are $J_{ev}(H_2O)$ at t = $t_{He}$ at the end of HCl evaporation. The colored circles in the data field mark the value of $J_{ev}(H_2O)$ after HCl deposition at t = $t_0$ and are equal to $J_{ev}(H_2O)$ of pure ice. The start of any particular $J_{ev}(H_2O)$ curve as a continuous solid (bold) line occurs at t = $t_D$ at 85% of $J_{ev}(H_2O)$ at t = 0 (pure ice value, colored circle or circled number on axis "b" to the left) and ends at $t_{Hb}$, the beginning of HCl evaporation as displayed in Figure 1B.

Figure 3: Synopsis of the dependence of the evaporation range parameter $r^{b/e}$ on the rate of deposition $R_{HCl}$ of HCl for temperatures between 188 and 210 K. Each point is marked with the total number of HCl molecules ($N_{HCl}$) deposited on the ice film , the temperature of the ice film at HCl deposition and the experiment number (bold) referring to Table 2. The hashed area encompasses $r^{b/e}$ values for dataset B (experiments 3,4,7,8). The color code goes from low (blue) over medium (green) to high (red) temperatures.

Figure 4: Synopsis of the dependence of $d_D$ on the rate of deposition $R_{HCl}$ of HCl for temperatures between 188 and 210 K. Each point is marked with the total number of HCl molecules ($N_{HCl}$) deposited on the ice film, the temperature of the ice film at HCl deposition and the experiment number (bold font) referring to Table 2. The hashed area encompasses $d_D$ values for dataset B (experiments 3,4,7,8). The color code goes from low (blue) over medium (green) to high (red) temperatures.

Figure 1

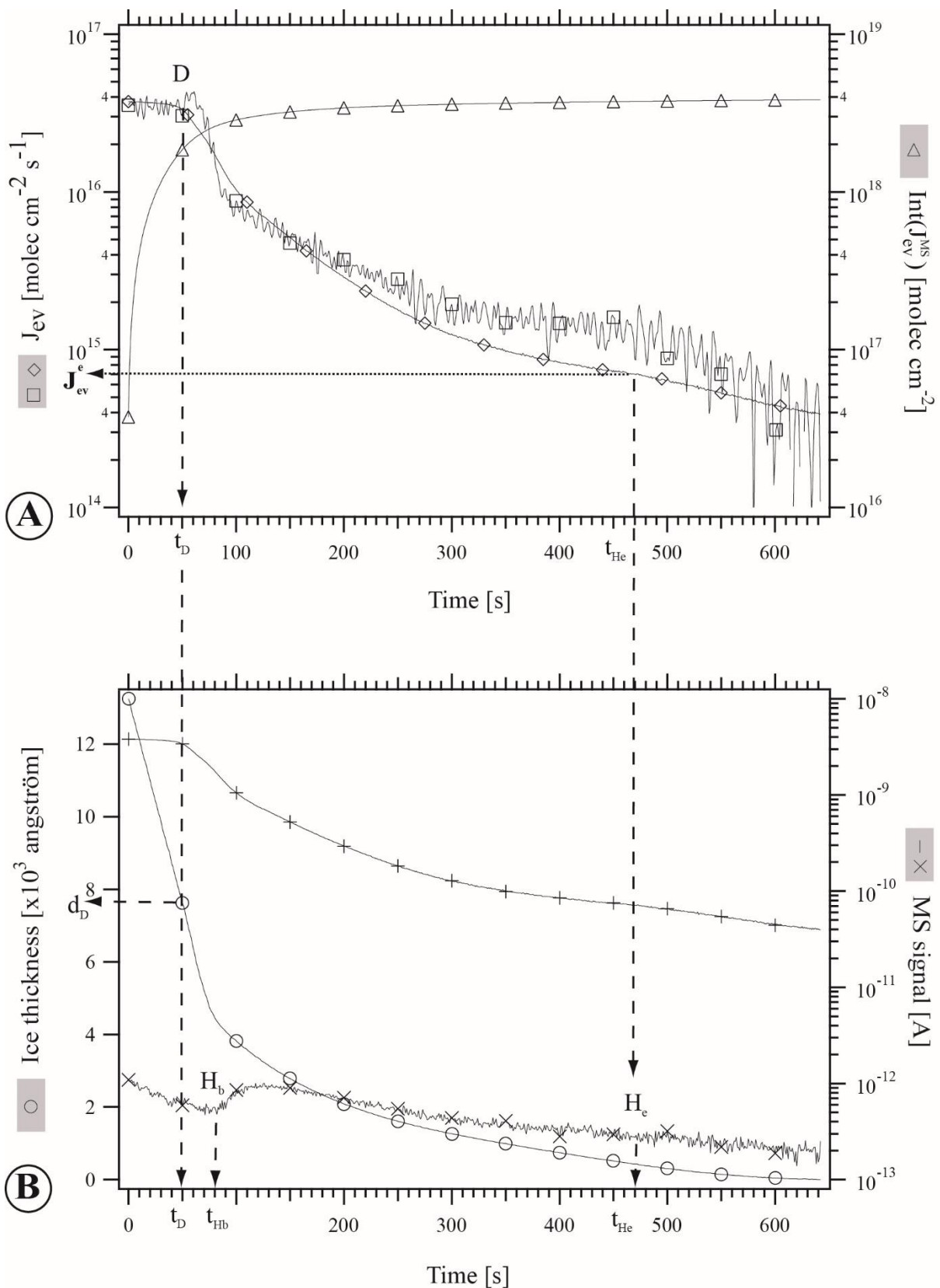

Figure 2

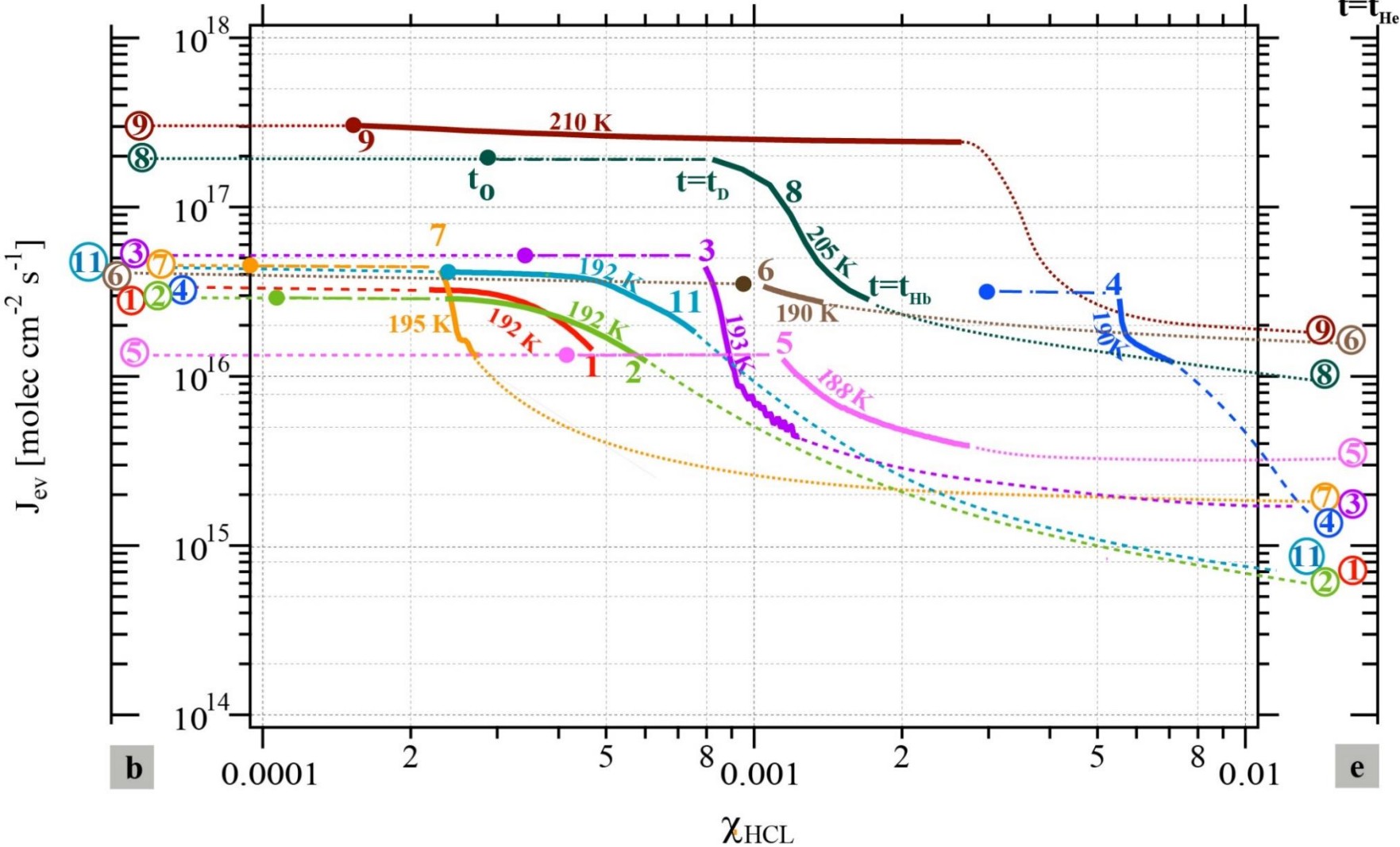

Figure 3

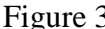

Figure 4

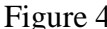

**APPENDIX A: Figures A1, A2, A3 and A4, Table A1**

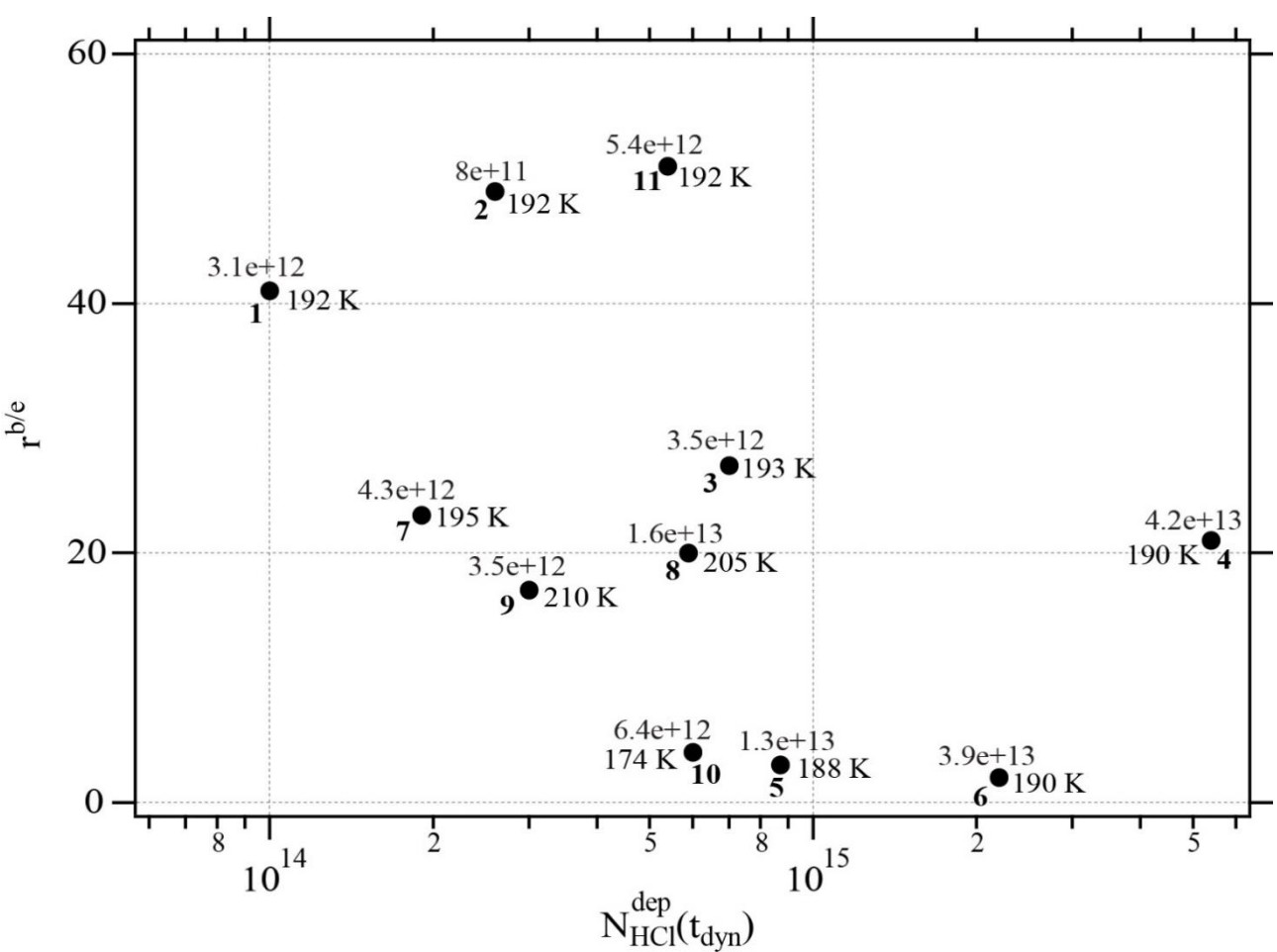

Figure A1: Synopsis of the dependence of the evaporation range parameter, $r^{b/e}$, on the number of adsorbed HCl, $N_{HCl}^{dep}$, adsorbed on ice for temperatures between 188 and 210 K. each point is marked with the deposition rate of HCl molecules in molec $s^{-1}$ on the ice film, the temperature of the ice film and the experiment running number (bold) referring to Table 2.

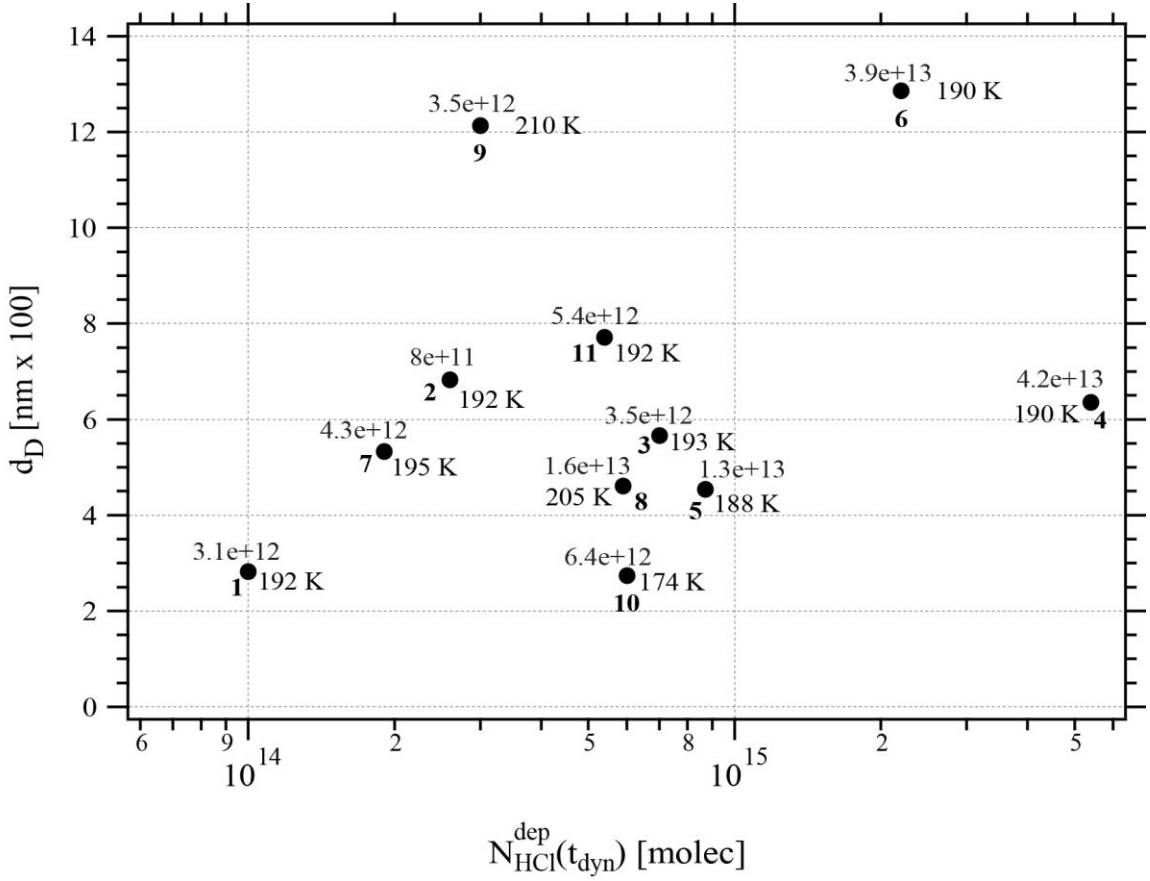

Figure A2: Synopsis of the dependence of the remaining thickness $d_D$ on the number of adsorbed HCl, $N_{HCl}^{dep}$, dispensed on ice for temperatures between 188 and 210 K. Each point is marked with the deposition rate of HCl in molec $s^{-1}$ on the ice film, the temperature of the ice film and the experiment running number (bold) referring to Table 2.

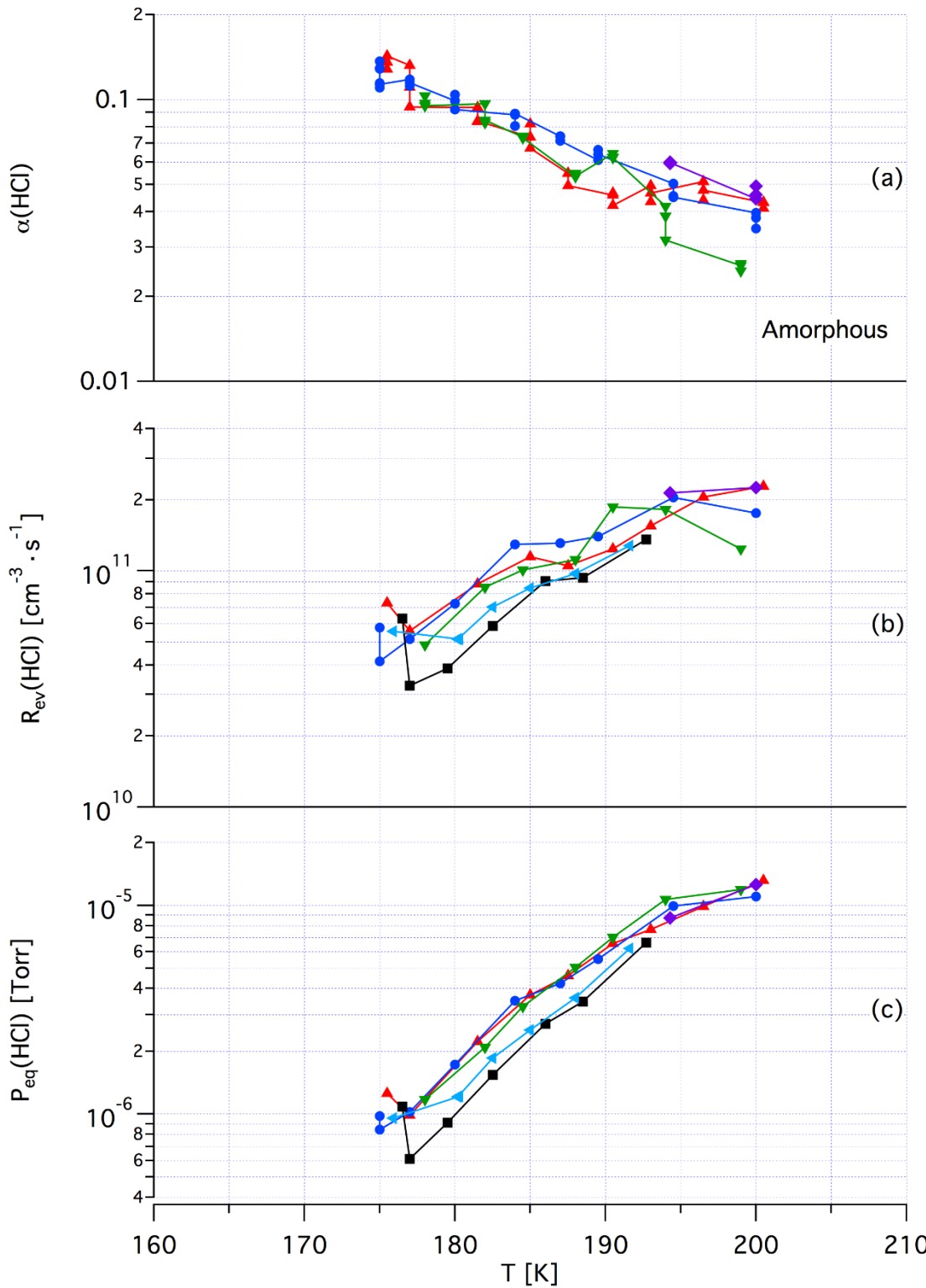

Figure A3: Synopsis of kinetic and thermodynamic results for an amorphous H₂O/HCl mixture using HCl as a probe gas. The symbols/colors used correspond to different experimental runs and the graphs show the scatter of the individual measurements within a series. Original data are published in Iannarelli, R. and Rossi, M.J. Atmos. Chem. Phys. 14, 5183–5204, 2014.

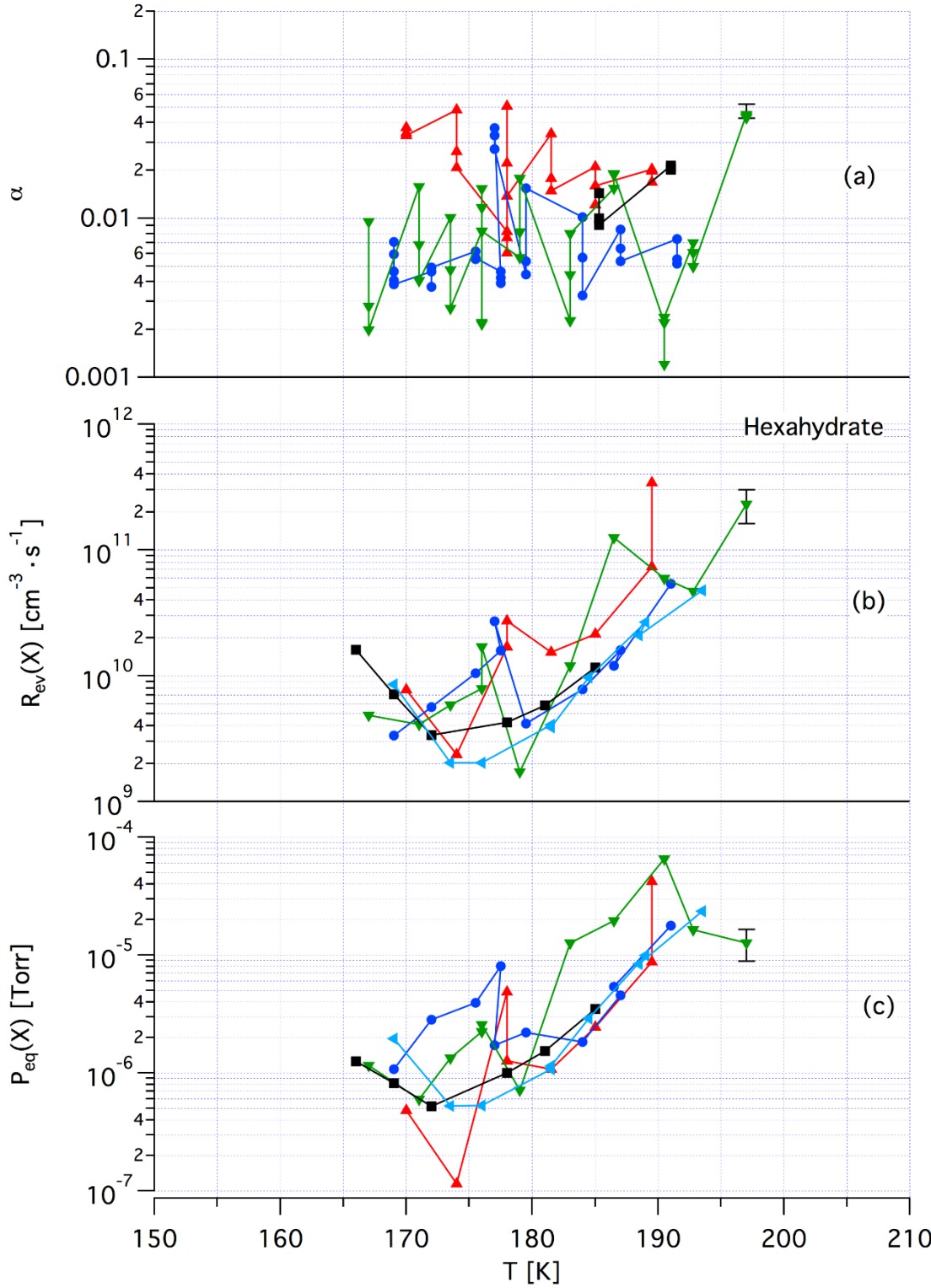

Figure A4: Synopsis of kinetic and thermodynamic results for crystalline HCl hexahydrate (HH) using X = HCl as a probe gas. The symbols/colors used correspond to different experimental runs and the graphs show the scatter of the individual measurements within a series. Original data are published in Iannarelli, R. and Rossi, M.J. Atmos. Chem. Phys. 14, 5183–5204, 2014.

## Table A1: Brief Summary of the Amount of a Molecular Monolayer (Coverage) of HCl adsorbed on H₂O ice

| Coverage / (molecule cm$^{-2}$) | Temperature / K | Bibliographic Reference |
|---|---|---|
| $5.0 \times 10^{15}$ | 200 | Hanson, D. R., Mauersberger, K., HCl/H₂O Solid Phase Vapor Pressures and HCl Solubility in Ice, J. Phys. Chem., 94, 4700–4705, 1990 |
| $1.0 \times 10^{15}$ | 200 | Abbatt, J.P.D., Beyer, K. D., Fucaloro, A. F., McMahon, J. R., Wooldridge, P. J., Zhang, R., Molina, M. J., Interaction of HCl vapor with water ice: implications for the stratosphere, J. Geophys. Res., 97, 15819–15826, 1992 |
| $(2.0 - 3.0) \times 10^{14}$ | 191 | Hanson, D., Ravishankara, A.R., Investigation of the Reactive and Nonreactive Processes Involving ClON0₂ and HCI on Water and Nitric Acid Doped Ice, J. Phys. Chem. 96, 2682-2691, 1992 |
| $1.15 \times 10^{15}$ | 183 | Foster, K. L., Tolbert, M. A., George, S. M., Interaction of HCl with Ice: Investigation of the Predicted Trihydrate, Hexahydrate, and Monolayer Regimes, J. Phys. Chem. A, 101, 4979–4986, 1997 |
| $2.5 \times 10^{14}$ | 208 | Interaction of HNO₃ with water-ice surface at temperatures of the free troposphere, Abbatt, J.P.D., Geophys Res. Lett. 24, 1479-1482, 1997 |
| $3.1 \times 10^{14}$ | 185 | Flückiger, B., Thielmann, A., Gutzwiller, L., Rossi, M. J., Real time kinetics and thermochemistry of the uptake of HCl, HBr and HI on water ice in the temperature range 190 to 210 K, Ber. Bunsenges. Phys. Chem., 102, 915–928, 1998 |
| $(1.1 \pm 0.6) \times 10^{14}$ | 201 | Lee, S.-H., Leard, D. C., Zhang, R., Molina, L. T., Molina, M. J., The HCl + ClONO₂ reaction on various water ice surfaces, Chem. Phys. Lett. 315, 7–11, 1999 |
| $(2.0 \pm 0.7) \times 10^{14}$ | 2001 | Hynes, R. G., Mössinger, J. C., Cox, R. A.: The interaction of HCl with water-ice at tropospheric temperatures, Geophys. Res. Lett. 28, 2827–2830, 2001 |
| $1.7 \times 10^{14}$<br>$1.3 \times 10^{14}$<br>$6.7 \times 10^{13}$ | 190<br>200<br>210 | Flückiger, B., Rossi, M.J., Common Precursor Mechanism for the Heterogeneous Reaction of D₂O, HCl, HBr, and HOBr with Water Ice in the Range 170-230 K: Mass Accommodation Coefficients on Ice, J. Phys. Chem. A 197, 4103-4115, 2003 |
| 2.3 to $2.7 \times 10^{14}$ | 180 to 200 | Henson, B. F., Wilson, K. R., Robinson, J. M., Noble, C. A., Casson, J. L., Worsnop, D. R.: Experimental isotherms of HCl and H₂O ice under stratospheric conditions, Connections between bulk and interfacial thermodynamics, J. Chem. Phys., 121, 8486– 8499, 2004 |