# Peer review of "The Influence of HCl on the Evaporation Rates of $H_2O$ over Water Ice in the Range 188 to 210 K at small Average Concentrations"

_Atmospheric Chemistry and Physics, 2018_

## Referee Comment (RC1) · Anonymous Referee #1 · 23 May 2018

The work "The Influence of HCl on the Evaporation Rates of H2O over Water Ice in the Range 188 to 210 K at small Average Concentrations"by Delval et al. reports the evaporative flux of water from ice/HCl mixtures at 170-210K. The experimental approach builds on the long tradition of excellently received papers from this group and the data are carefully analysed and I have no doubt that the result statements are well supported by the data. Further, the topic addresses core physical chemistry of ice with its importance for atmospheric science. Therefore, the topic fits perfectly into ACP and I'd support publication after a revision of the manuscript. I'm sorry to reject it in its current form.

[Figure]

The main reasons for asking for a major revision are

o Limited Discussion: The manuscript tends to stop at the level of reporting the results without relating them to the results by other groups or lifting them to a more general level.

o No Relevance Given: The introduction is very interesting to read and reveals a detailed discussion on key-topics relevant to the ice-HCl system. However, questions key to this study are not covered:

+ Why do we need to know J(des)?

+ Where and when is the lifetime of ice particles critical and is the water flux the determining factor?

+ How relevant are the non-equilibrium desorption processes described here to the environment? Please, do not get me wrong. I do believe this lists topics that are nicely addressed by this study and are highly relevant to the environment. It is primalrey the question of discussing those in the text.

o Structure: For my feeling, the manuscript jumps to much back and forth between the topics. It is rather difficult to follow.

I'd kindly ask you to address these issues and would welcome a revised version. In the following, I give some detailed questions that aim at guiding you. This is not a complete list, and I kindly ask to address the major topics first. A new review can then address the details.

Detailed comments:

Introduction, p2: The molecular and dynamic details of crystallization are mentioned. Could you give details on what this would mean for your experiment. What the role of eventually slow formation dynamics in the preparation of your samples, where apparently you start with pure ice to which to dope HCl.

[Figure]

Introduction, p3: Where is the paragraph starting with Fourier-Transform IR heading? What is the take home message with respect to your work?

Introduction, p3: "Regarding the nature of HCl-ice adsorbate, " What has ionisation to do with your study? This is a long and detailed description in the introduction to which you never return in the discussion.

Experimental: Please specify how did you quantify HCl? How did you derive the mole fraction, i.e. how did you get the volume of ice? Did you assume homogeneous mixture in the total volume of ice? Why is that appropriate? Could you specify on mixing and diffusion times? What is the error on the mole fraction?

Results, p 7 The average mole fraction should be called "apparent"?

Discussion:Please add discussion of other work on H2O Fluxes from ice in presence of acidic gases. Can your findings be related to water fluxes from other surfaces? Is this result part of a larger picture?

––––––––––––––––––––––––

---

## Referee Comment (RC2) · J. P. Devlin (Referee) · 12 Jun 2018

(Throughout we will recognize references from Duval and Rossi using the first name with an asterisk)

Having given the authors rather high rankings and favored the publication of their manuscript, I will now concentrate on a few aspects where some weaknesses need attention. In particular, I do not feel nearly as congratulatory toward the group of Laffon and Parent as the authors appear to be on page 5. The new data and some of the insights from the Parent group may be useful, but it is difficult to be sure when they are careless in other respects; for example, refusing to expand their data where there

are clearly uncertainties; even when a useful expansion can be had quite simply, as by replacing H by D (see later). For that reason in 2011 I collaborated with Heon Kang to produce a 2011 Comment (PCCP 06-2011-022007) on their 2011 publication, i.e. "Comment on "HCl adsorption on ice at low temperature: a combined X-ray absorption, photoemission and infrared study" P. Parent, J. Lasne, G. Marcotte and C. Laffon, Phys. Chem. Chem. Phys., 2011, 13, 7142) and will use our "Comment" as a partial base for points made here in light of the praise on page 5. I will focus on the comparison of the infrared data and assignments from our lab with that of Parent's. The insights of Kang with respect to the comparative data from other methods can be found in our earlier Comment noted above. This is not easy as in the new 2011 paper Parent quietly makes a complete assignment reversal on what we have always regarded as the major band, a strong one at 2480 cm-1 that was 1st viewed and assigned as singly-hydrated molecular HCl in our 2002 Nature paper. This band first emerges at 2480 but shifts to 2520 with warming from 50 K to above 60 K or the addition of HCl beyond $\sim$20% of an ML. Parent has never acknowledge a second band or this shift, but in 2011 changes the identity of the "single band", for all temperatures and dosages, from what he had regarded as an 100% ionized HCl band to a single band assigned strictly to molecular HCl. Had he vertically expanded the 50 K 0.15 ML figure near the bottom in Fig.4 he may have been forced to accept that the band starts below 2500. The assignments of most other significant bands were left unchanged; in particular retaining 1740 as assigned as hydronium ion rather than our long-term assignment as doubly hydrated molecular HCl.

In an aside from the older data, it may be helpful to realize that we now have spectra for HX/DX molecules bound to oxygen of H2O on the small-cage walls of gas clathrate hydrates. Bonding to this oxygen becomes available when an ether in a neighbor cage steals a hydrogen from an oxygen of the intervening wall. These relatively new (2015) data in Fig.1 have been made accessible through the recent (2010) development of an all-vapor premixture methodology that commonly reduces the clathrate formation process from hours to sub-seconds. Paired with the corresponding adsorbate case

for the ice surface we find: HCl – 2482/2480; DCl – 1880/1870; HBr - 2350/2300. These are all broad bands with some manageable interference from other bands. The published spectra are available in the literature for all six cases; for discussion of the three clathrate spectra see DOI: 10.1021/acs.jpca.5b07019; J. Phys. Chem. A 2015, 119, 9018−9026.The close comparisons of the clathrate-wall and ice-surface values of bonded HX were surprising, but they support the new 50 K molecular assignment of Parent as well as our continuing assignment of the 2480 ice-surface band to molecular HCl. Most importantly they seem to show conclusively that one must expect a band near 2500 for molecular HCl with a single-bond to surface oxygen. Parent assumes that and assigns the 2500 band, the strongest band in the spectrum at 50 K and 0.5 ML, to the molecular HCl. Then he goes fully off-track by a) first suggesting that continued growth at 2500 reflects increasing single-bonded molecular HCl and then near the end of the paper criticizing Kang and myself for suggesting the concept of a very limited amount (0 -15%) of ionized HCl for low-dosages at 50 K. This seems to have been a required statement flowing from his insistence that there is only a single band near 2500. However, it fully inverts the generally accepted view that molecular HCl is prominent at 50 K while ionic versions, including the Zundel ion, dominate at higher temperatures /higher dosages. Keep in mind that we do report, in the 1998 paper of Uras*, a figure showing molecular DCl on ice at 125 K close to a much stronger ion band. The 2520 ion band (our assignment) does weaken at higher temperature as the ions switch to Zundel forms (see the top of Figure 2 of Devlin*).

Now let's focus on the 1740/1710 band that is agreed to be present in all cases below 90 K for doses less than 0.15 ML Here we can make a simple molecular switch, one of several choices that we have shown in Buch* and Devlin*, to attack the Parent assignment as an ionic band in all cases. That switch is to retain HCl while replacing H2O by D2O as in the bottom section of Fig. 1 of Devlin*. The figure-caption states correctly that "The use of D2O ice also eliminates possible contamination of this band by protonated-water absorption" since the ionic band should downshift by a factor of ∼1.36 because of mass and anharmonic effects. The figure shows that our 1710 band

is largely unaffected by the isotopic switch so that it cannot be exclusively an ionic band. This apparently eliminates the Parent-group strictly ionic assignment; but lets' consider an all-deuteration switch to also eliminate the very unlikely possibility that a different ion-band shifted into the 1740 cm-1 position. With complete deuteration, we can predict the shifted HCl band position by dividing the molecular 2480 band frequency by 1.36, obtaining 1824. Figure 2 of Devlin* shows the band is in fact at 1870. This seems to be an acceptable match that would be even closer (1853) had we used the Parent new molecular-value of ∼2520. So I stop here rather than going on to a comparison with data for HBr as in Buch*, Fig.9 which also works quite well in establishing the one and two HX coordinations with the ice surface at low temperatures and dosages.

Conclusions: The Parent group data is mostly reliable but the interpretations are not. The 50 K low dosage spectra clearly include bands for both single (2480) and doubly (1710) hydrated molecular HCl. Clathrate-hydrate data for HX and DX are a surprising source of relevant data. There seems no basis to reject band assignments from 2002 (Buch* and Devlin*).

1. Perhaps it should be clarified that RAIR FTIR data often cannot be compared directly with absorbance data. 2. This current review of the data convinces me that at least between 40 and 130 K the Bartels-Rausch recent HCl-on-ice publication may be right about the presence of molecular HCL on ice surfaces to some extent at most temperatures (but not all conditions). Is there a reason that paper was not included in the review? 3. The Zundel ion is important in understanding the evolution of the changes that follow warming after HCl has been adsorbed on ice at low levels at low T. Also the basic characteristics of the Zundel ion vs its environment have been thoroughly examined in the last 10 years. I particularly recommend an older joint review paper of Buch, Parrinello and others on the hydrates of HCl; J. Phys. Chem. A 2008, 112, 2144-2161 that highlights the Zundel ion. 4. On page 5 the authors question the importance of the dangling O=H bonds during the early uptake of HCl at low temperature. Is this consistent with the Parent infrared data in their 2011 paper in which they show/remark that

the dH-band is gone after 0.15 ML at 50 K. Why did you make the statement WITHOUT GIVING A REASON?

Fig,1 HX spectra in clathrate hydrates.

[Figure]

**Fig. 1.** Fig,1 HX spectra in clathrate hydrates.

---

## Author Comment (AC1) · 3 Aug 2018

**Answer to referees regarding "The Influence of HCl on the Evaporation Rates of H2O over Water Ice in the Range 188 to 210 K at small Average Concentrations by C. Delval and M.J. Rossi, Atmospheric Chemistry and Physics Discussions (https://doi.org/10.5194/acp-2018-238-RC2, and -RC1, 2018)**

We would like to thank both referees for their comments, questions and remarks. We have taken into account all questions and comments of both referees to various degrees of implementation in the revised version of above-referenced paper thereby wanting to avoid unnecessary duplication of text in relation to previous papers dealing with the same subject. We have outlined our actions on the manuscript point-by-point and will highlight the added/modified text in "track change" format once we will have the encouragement for submission of the revised version of the manuscript by the acp editors. We report the questions/comments of the referees in *italics* followed by our answer in straight font.

**Answer to Dr. J.P. Devlin (referee 2):**

We would like to address our sincere thanks to this referee for pointing out the inconsistencies in the assignment of the IR reflection-absorption (RAIR) spectra in the work of Parent and coworkers as stated in your comment. We did indeed miss these author's reassignment in their recent 2011 work and are grateful for your astute observations. We cannot contribute much to confirm or deny their assignment from our own chemical kinetic work, but like to agree with you that the conversion usually follows the route "molecular" to "ionic" with increasing temperature and not the inverse, akin to the results on N2O5 deposition on ice films in the seminal work of John Sodeau and Andy Horn some time ago (J. Phys. Chem. 98, 946-951, 1994). However, what we may contribute are some shortcomings in the Parent and coworker's study of 2011 (Parent et al., Phys. Chem. Phys. 13, 7142-7148, 2011) from our point of view:

- Their value for a monolayer is a factor of approximately four too high compared to the consensus value of 2-3 1014 molecule cm-2. Although most of the early work resulted in distinctly larger values, probably owing to the phase transition to amorphous, the newer studies under stricter experimental control all settle around the indicated value according to Table 1 below. Although not explicitly stated, the conversion of their exposure to coverage values makes use of 8.0 1014 molecule cm-2 as a molecular monolayer. Using a factor of four lower value surely will affect the calculated surface concentrations.
- Somewhat related, the derived concentrations in Figure 3 (XPS, 50 and 90 K concentrations) and Figure 4 (FT-RAIR) do not have any uncertainty limits which has apparently slipped the attention of the competent editor! The reader does not have a clue regarding uncertainties and sensitivities/limits of detection.
- The deposition of CRYSTALLINE Ice at 150 K (for the NEXAFS experiment) is doubtful! In order to make sure that one has indeed Ih ice one must anneal the deposit up to 190 K in order to be sure of the structure and subsequently cool down! What they most probably have is a disordered Ih ice phase that mimicks the often sought but so far elusive atmospheric cubic ice phase (Ic), probably together with amorphous domains (see work by W. Kuhs and co-workers).
- The authors deposit 4 ML's of amorphous ice at 110-120 K for XPS, UPS and FT-RAIR and compare it to 100 ML's amorphous ice deposited at 150 K for NEXAFS experiments. Most likely, the density of the "amorphous ice" is different following astrochemical work (high density vs. low density amorphous). Even though the species observed are qualitatively similar in both environments, the question of comparability of results must be posed. As long as it cannot be

shown that the transformation/conversion rates are not identical in both ice films no statement about relative abundances may be made. IR or RAIR is probably not an ideal monitoring technique for radiation damage of a sample (disparate beam diameters!). It seems to us that Parent and coworkers cannot assert that the ice substrate is identical for all spectroscopic experiments reported, and this will have consequences for HCI deposition.

- We are critical of the "dangling" H (dH) results in Figure 4 in terms of "negative" IR absorption peaks. Following our own unpublished work dH disappears at 220 K both in the presence and absence of HCl without apparent kinetics/reactivity at lower temperature that would manifest itself as a decrease of dH abundance upon HCl exposure. We reckon that the abundance of these sites is low enough such that HCl may interact with other sites on the existing ice surface. We note the absence of a correlation between the decrease of the number of dH sites with the appearance of "new" IR absorptions up to 220 K.
- Comparing the 50 K results in Figure 3 from 0 to 3 L exposure with the calculated concentrations in mol % (right ordinate) and coverage in ML (left ordinate), it is not obvious to us how these results were derived and how sensitive they are to calibration (see above under HCl monolayers). Uncertainty limits would be quite helpful in this instance, especially so as these data are the cornerstone of their conclusions. The same goes for the correspondence between exposure and concentrations displayed in Table S1.

As a result we will briefly summarize the situation in the introduction of the manuscript in the revised version. By the same token we will attenuate the significance of the work of Parent et al. in light of the obvious shortcomings pointed out above and made worse by the recent reversal of vibrational assignments occurred in their most recent (2011) compared to previous (2005) work.

| Value / (molecule cm -2 ) | Temperature / K | Bibliographic Reference                                                               |
|--------------------------------------|-----------------|---------------------------------------------------------------------------------------|
| 5.0 10 15                 | 200             | Hanson, D. R., Mauersberger, K., HCl/H 2 O Solid Phase Vapor Pressures and |
|                                      |                 | HCl Solubility in Ice, J. Phys. Chem., 94, 4700–4705, 1990                            |
| 1.0 1015                             | 200             | Abbatt, J.P.D., Beyer, K. D., Fucaloro, A. F., McMahon, J. R., Wooldridge, P.         |
|                                      |                 | J., Zhang, R., Molina, M. J., Interaction of HCl vapor with water ice:                |
|                                      |                 | implications for the stratosphere, J. Geophys. Res., 97, 15819–15826, 1992            |
| (2.0 – 3.0) 10 14         | 191             | Hanson, D., Ravishankara, A.R., Investigation of the Reactive and                     |
|                                      |                 | Nonreactive Processes Involving CIONO 2 and HCI on Water and Nitric Acid   |
|                                      |                 | Doped Ice, J. Phys. Chem. 96, 2682-2691, 1992                                         |
| (7.3 ± 0.5) 10 15         | 183             | Foster, K. L., Tolbert, M. A., George, S. M., Interaction of HCl with Ice:            |
|                                      |                 | Investigation of the Predicted Trihydrate, Hexahydrate, and Monolayer                 |
|                                      |                 | Regimes, J. Phys. Chem. A, 101, 4979–4986, 1997                                       |
| 2.5 10 14                 | 208             | Interaction of HNO 3 with water-ice surface at temperatures of the free    |
|                                      |                 | troposphere, Abbatt, J.P.D., Geophys Res. Lett. 24, 1479-1482, 1997                   |
| 3.1 10 14                 | 185             | Flückiger, B., Thielmann, A., Gutzwiller, L., Rossi, M. J., Real time kinetics        |
|                                      |                 | and thermochemistry of the uptake of HCl, HBr and HI on water ice in the              |
|                                      |                 | temperature range 190 to 210 K, Ber. Bunsenges. Phys. Chem., 102, 915–                |
|                                      |                 | 928, 1998                                                                             |
| $(1.1 \pm 0.6) \ 10^{14}$            | 201             | Lee, SH., Leard, D. C., Zhang, R., Molina, L. T., Molina, M. J., The HCI +            |
|                                      |                 | CIONO 2 reaction on various water ice surfaces, Chem. Phys. Lett. 315, 7–  |
|                                      | 2004            | 11, 1999                                                                              |
| $(2.0 \pm 0.7) \ 10^{14}$            | 2001            | Hynes, R. G., Mossinger, J. C., Cox, R. A.: The Interaction of HCI with water-        |
| 4 7 4 0 1 4                          | 100             | Ice at tropospheric temperatures, Geophys. Res. Lett. 28, 2827–2830, 2001             |
| 1.7 1014                             | 190             | Fluckiger, B., Rossi, M.J., Common Precursor Mechanism for the                        |
| 1.3 1014                             | 200             | Reterogeneous Reaction of D 2 O, HCI, HBr, and HOBr with Water ice in the  |
| 6.7 10 13                 | 210             | A 107 4102 4115 2002                                                                  |
| 2 2 4 2 7 1014                       | 100 to 200      | A 137, 4103-4113, 2003                                                                |
| $2.3 \text{ to } 2.7  10^{14}$       | 180 to 200      | Mershon, D. P.; Evnerimental isotherms of HCl and H-Q iso under                       |
|                                      |                 | stratespheric conditions. Connections between bulk and interfacial                    |
|                                      |                 | thermodynamics I Cham Phys. 121 8486–8400 2004                                        |
|                                      | 1               | ulerinouynamics, J. Chem. Phys., 121, 0400 – 0433, 2004                    |

Table 1: Brief Summary of the Amount of a Molecular Monolayer of HCl adsorbed on H2O ice

In summary, the work of Parent et al. (2011) glosses over the properties of the ice substrate itself that may be different for different experiments, such as NEXAFS, XPS, UPS and RAIR spectroscopies. It behooves the authors to search for a relevant experimental marker that pinpoints the properties of the ices investigated as a function of temperature and mode of deposition. There is room for improvement.

The additional questions posed by the referee may be answered in the following:

1. Perhaps it should be clarified that RAIR FTIR data often cannot be compared directly with absorbance data

The discrepancy between RAIR and FTIR absorption data will be pointed out in the main text although the interfacial character of the obtained data is emphasized in polarized (s and p-type) RAIR data which is desirable if you are interested in the composition of the interface. In all honesty, we would have liked to record our recent NAT and NAD data in polarized RAIR fashion if it were not for the missing capability to measure quantitative absorption cross sections (R. lannarelli and M.J. Rossi, J. Geophys. Res. Atmos., 120, 11,707–11,727, doi:10.1002/2015JD023903).

2. This current review of the data convinces me that at least between 40 and 130 K the Bartels-Rausch recent HCl-on-ice publication may be right about the presence of molecular HCl on ice surfaces to some extent at most temperatures (but not all conditions). Is there a reason that paper was not included?

The Bartels-Rausch paper on HCl-ice was not included in the first place because the evidence the authors present is not ultra-strong. However, for completeness sake, we will list it in the bibliographic references being a newer piece of work recorded using a synchrotron.

3. The Zundel ion is important in understanding the evolution of the changes that follow warming after HCl has been adsorbed on ice at low levels at low T. Also the basic characteristics of the Zundel ion vs its environment have been thoroughly examined in the last 10 years. I particularly recommend an older joint review paper of Buch, Parrinello and others on the hydrates of HCl; J. Phys. Chem. A 2008, 112, 2144-2161 that highlights the Zundel ion.

We will make mention of the importance of the Zundel IR absorption continuum in the revised text in relation to its temperature dependence. The reason we did not consider it was that the present study focused on higher temperatures and on thermochemical kinetic aspects rather than on IR vibrational assignments.

4. On page 5 the authors question the importance of the dangling O-H bonds during the early uptake of HCl at low temperature. Is this consistent with the Parent infrared data in their 2011 paper in which they show/remark that the dH-band is gone after 0.15 ML at 50 K. Why did you make the statement WITHOUT GIVING A REASON?

We refer to the point raised above in relation to the dangling (isolated) –OH bond. In our hands the behavior of the dH bonds was different, most probably owing to the higher temperatures used. We do not want to get involved in discussions in view of our unpublished results at variance with Parent et al. (2011).

**Answer to referee 1:**

Most of the points raised by referee 1 have already been mentioned in the original text. However, the referee questions/comments show us where we could have been clearer in our explanations, and we nevertheless thank the referee for his diligence and attention to detail. We will emphasize or complete our original text, as the case may be. Here, we will give brief answers to the specific questions raised by the referee.

o Limited Discussion: The manuscript tends to stop at the level of reporting the results without relating them to the results by other groups or lifting them to a more general level

There are simply no other measurements of absolute desorption rates of molecules constituting the components of contaminated ices of atmospheric relevance. All existing reports on desorption rates  $J_{des}(M)$  of  $H_2O$ , HCl, HNO3, adsorbed organics and the like make the assumption of unity accommodation coefficients which may be wrong by up to three orders of magnitude, whereas the present results were obtained without the incidence of readsorption owing to the small absolute pressures used. The only other study with which to compare the results of HCl is our own dealing with HNO3 absolute rate of desorption referenced in the bibliography (Delval and Rossi, 2005). We will therefore emphasize this point in the discussion section.

o No Relevance Given: The introduction is very interesting to read and reveals a detailed discussion on key-topics relevant to the ice-HCl system. However, questions key to this study are not covered: + Why do we need to know J(des)?

The value of  $J_{ev}(H_2O)$  ( $J_{des}(H_2O)$ ) is directly related to the net evaporative lifetime once the relative humidity (rh in %) is known, or in the present case, to lifetime prolongation of contaminated atmospheric ice particles with respect to pure water ice. The reciprocal value  $1/J_{ev}(H_2O)(100\text{-rh})$  scales with the time it takes to evaporate a given mass of water ice at a given temperature and rh value. As pointed out above, there are no measured lifetimes of ice particles except those based on vapor pressures in conjunction with a unity accommodation coefficient that lead to significantly shortened lifetimes of ice particles.

**+ Where and when is the lifetime of ice particles critical and is the water flux the determining factor?**

Lifetimes are important when gauging heterogeneous chemistry on ice particles with reactive "reservoir" species in the UT/LS region of the atmosphere as is the case in polar stratospheric chemistry (ozone hole), polar boundary layer ozone disappearance (so-called bromine explosions) or global heterogeneous ozone disappearance on sulfate aerosols in the UT/LS region of the atmosphere. This region of the atmosphere is subject to frequent under- and sometimes oversaturation because the stratosphere is often "dry", that is undersaturated in water vapor.

+ How relevant are the non-equilibrium desorption processes described here to the environment? Please, do not get me wrong. I do believe this lists topics that are nicely addressed by this study and are highly relevant to the environment. It is primarily the question of discussing those in the text.

You can easily convince yourself of the variability of water vapor saturation by looking at contrails that sometimes persist only for a second or so before evaporation (atmosphere heavily undersaturated) or persistent for many hours (atmosphere close to equilibrium w/r to water vapor pressure). What is relatively new is the fact that the UT/LS region is finely structured in that strata (atmospheric layers)

of only a few hundred meters thickness change between undersaturation and saturation in water vapor. The balloon-sonde measurements of Terry Deshler some time ago revealed this fine structure in a clear manner. We are refraining from going into meteorology in the present paper that deals with fundamental physical chemistry of evaporation of H2O and HCl from model atmospheric ice particles.

o Structure: For my feeling, the manuscript jumps to much back and forth between the topics. It is rather difficult to follow. I'd kindly ask you to address these issues and would welcome a revised version. In the following, I give some detailed questions that aim at guiding you. This is not a complete list, and I kindly ask to address the major topics first. A new review can then address the details.

Here we take exception to the first statement made by referee 1. We cannot and will not rearrange the structure of the paper without more specific advice from this referee. I can assure the referee that much thought has gone into the planning and structuring of the present paper.

**Detailed comments:**

Introduction, p2: The molecular and dynamic details of crystallization are mentioned. Could you give details on what this would mean for your experiment. What the role of eventually slow formation dynamics in the preparation of your samples, where apparently you start with pure ice to which to dope HCl.

The structural nature of the ice substrate has some incidence on the chemical kinetics of adsorption/desorption of polar molecules as documented in Figure A3 and A4. On a very fundamental base different pure water ices have different values of  $J_{ev}(H_2O)$  and water accommodation coefficient  $\alpha(H_2O)$ . This has been published in the literature many times, and also by us (see for instance Pratte et al., 2006). See also our comments above in response to referee 2. However, the referee is correct in pointing out this weakness from which many published studies suffer and which should be the starting point of more careful investigations in the laboratory. It turns out that the molecular structure and morphology of the ice generated in the laboratory is very important to adsorption/desorption processes, and this has not always been recognized.

Introduction, p3: Where is the paragraph starting with Fourier-Transform IR heading? What is the take home message with respect to your work?

Regarding the first part of the question we have no clue what the reviewer is referring to. For the takehome lesson please refer to the "Conclusions" section at the end of the manuscript.

Introduction, p3: "Regarding the nature of HCI-ice adsorbate," What has ionisation to do with your study? This is a long and detailed description in the introduction to which you never return in the discussion.

"Ionisation" is not at stake here, it is rather electrolytic dissociation that is at the centerpoint of the discussion on HCl-contaminated ices. The goal here is to introduce the reader to a problem in the description of the HCl/ice interface that has motivated countless researchers in the past. We do not mention this in the discussion because we have nothing to contribute to this topic in the present study.

Experimental: Please specify how did you quantify HCl? How did you derive the mole fraction, i.e. how did you get the volume of ice? Did you assume homogeneous mixture in the total volume of ice? Why is that appropriate?

For further experimental details please see Delval et al. (2005). The amount (number) of HCl molecules lost to the ice surface was measured at the outset upon exposure of the ice to HCl. After total evaporation of the ice film HCl was monitored quantitatively with a mass balance closed to 80% after recovery as a check. The mole fraction was determined from the number of moles of HCl and  $H_2O$ because both were monitored quantitatively, thus this method is independent of the volume of ice. As mentioned in the manuscript our method is unfortunately not able to determine concentration profiles such that we cannot comment on concentration gradients upon HCl doping. However, we have measured the self-diffusion coefficient of HCl and HBr in ice some time ago at low temperature as described in the work of Aguzzi et al. (Phys. Chem. Chem. Phys. 5, 4157-4169, 2003) and Flückiger et al. (J. Phys. Chem. 104, 11739-11750, 2000).

Could you specify on mixing and diffusion times? What is the error on the mole fraction? Results, p 7 The average mole fraction should be called "apparent"? Discussion: Please add discussion of other work on H2O Fluxes from ice in presence of acidic gases. Can your findings be related to water fluxes from other surfaces? Is this result part of a larger picture?

Regarding diffusion times see reference given above. The error/uncertainty on the mole fraction is estimated at 10%, mostly coming from the determination of the loss of HCl on the walls of the reaction vessel. We are opposed to calling the mole fraction "apparent" and will continue to use "average" because we would like to emphasize the average nature! As stated above with insistence there are no other studies dealing with H2O fluxes from water ice in the presence of acidic gases. It turns out that the behavior of  $J_{ev}(H_2O)$  from HCl-doped ice is complex which at this time precludes the transfer of knowledge to other substrates. A larger picture will certainly emerge once more studies on other aspects of evaporation/condensation will have performed.

---

## Author Response (AR1)

**Point-by-point Response to Referee Comments for manuscript acp-2018-238 "The Influence of HCl on the Evaporation Rates of H2O over Water Ice in the Range 188 to 210 K at small Average Concentrations" by Christophe Delval and Michel J. Rossi**

Enclosed please find the final author comments regarding above-mentioned manuscript. We are using the abbreviations of the manuscript (e.g. Parent et al., 2011) to call out specific references in the present response. In addition to the text listed here we have added a Table in the Appendix (Table A1) showing ten references for the quantity corresponding to a molecular monolayer of HCl adsorbed on ice. There are many more changes we have made to the manuscript compared to the listed textual changes in response to the referee's comments in order to clarify certain aspects mentioned by Referee 1. The references to lines refer to the annotated manuscript (track change).

**Referee 2 (Dr. J.P. Devlin)**

*Referee Comment*: We refrain here from reproducing the lengthy three-page comment by the referee. It is essentially a summary of discrepancies in FTIR assignments in the mid-IR range between the work performed by Parent et al. (2011) that we have overlooked, and his own including his associates.

*Author's Response*: We admit that we have overlooked the Parent et al. (2011) work in the original manuscript and the interesting comments that followed the 2011 Parent publication by Devlin and Kang (2012) and its response by Parent et al. (2012). We have added a § in the Introduction to that effect and have completed the bibliographic section in the manuscript accordingly.

Work by Parent and coworkers uses Near-Edge X-Ray Absorption Spectroscopy

Author's Changes to manuscript: This § has been inserted into the Introduction, lines 140 to 168.

(NEXAFS) of HCl-doped low temperature ice substrates in order to determine the relative population of ionic and covalently bound HCl and distinguish between bulk and HCl surface states in the temperature range 20 to 150 K (Bournel et al., 2002; Parent and Laffon, 2005). The results seem to confirm the consensus on the low-temperature existence of molecularly adsorbed HCl up to 90 K beyond which an increasing amount of HCl is converted into an ionic form, such as H3O+Cl- (Eigen cation) or H5O2+Cl- (Zundel cation) formed by spontaneous ionization of adsorbed HCl on ice, up to completion at 150 K. The newest work by Parent compares NEXAFS with photoemission (UPS, XPS) and FTIR in transmission of thin HCl/H2O films (Parent et al., 2011). The results are roughly consistent but surprising in the sense that these workers find 92% ionically dissolved HCl in/on ice at 50 K in contrast to Kang et al. (2000) and Devlin et al. (2000) under similar exposure (dose) and temperature conditions. In addition, Parent et al. (2011) perform the NEXAFS experiment on a (thick) 100 ML "crystalline" H2O ice substrate deposited at 150 K whereas the photoemission and FTIR experiments used a 4 ML thin ice slab deposited at 120 K. The question has to be raised whether the two types of used ice films may be responsible for some of the discrepancies in the results because both the density and the structure of ice are known to be a strong function of temperature and deposition conditions (Kuhs et al., 2012; Schriver-Mazzuoli et al., 2000). The most recent work of Parent et al. (2011) sparked an interesting controversy in the assignment of the FTIR absorption spectrum of thin HCl/H2O films and led to two comments showcasing the difficulties of intercomparison of nominally identical experiments (Devlin and Kang, 2012; Parent et al., 2012).

*Referee Comment:* 1. Perhaps it should be clarified that RAIR FTIR data often cannot be compared directly with absorbance data.

Author's Response: Agreed. Text inserted on lines 100 - 101.

Author's Changes to manuscript: It must be recalled that FTIR spectra in transmission and reflection may in most cases not be directly compared across the mid IR range.

*Referee Comment:* 2. This current review of the data convinces me that at least between 40 and 130 K the Bartels-Rausch recent HCI-on-ice publication may be right about the presence of molecular HCL on ice surfaces to some extent at most temperatures (but not all conditions). Is there a reason that paper was not included in the review?

*Author's Response*: We originally did not include the Bartels-Rausch reference because of technical shortcomings including possible uncertainties in the HCl profile using energy analysis of the photoelectrons and other, more quantitative questions. We now include this reference at the end of the Introduction section on lines 178-183 in order to accommodate the referee's suggestion.

*Author's Changes to manuscript*: The most recent experimental work on HCl/H2O at an atmospherically relevant ("warm") temperature (253 K) has examined the HCl depth profile using XPS spectroscopy and finds molecularly adsorbed (physisorbed) HCl at its outermost layer and ionic dissociation in deeper layers (Kong et al., 2017). Complementary X-Ray absorption results also point towards a perturbation of the crystal structure of ice in the aftermath of HCl adsorption/dissolution into deeper layers of ice.

*Referee Comment:* 3. The Zundel ion is important in understanding the evolution of the changes that follow warming after HCl has been adsorbed on ice at low levels at low T. Also the basic characteristics of the Zundel ion vs its environment have been thoroughly examined in the last 10 years. I particularly recommend an older joint review paper of Buch, Parrinello and others on the hydrates of HCl; J. Phys. Chem. A 2008, 112, 2144-2161 that highlights the Zundel ion.

*Author's Response*: We have briefly mentioned the Zundel ion on line 144 to 154 including the insertion of the relevant bibliographic reference. However, we have refrained from further elaboration owing to the focus of the present work which is kinetic rather than spectroscopic.

Author's Changes to manuscript: The results seem to confirm the consensus on the low-temperature existence of molecularly adsorbed HCl up to 90 K beyond which an increasing amount of HCl is converted into an ionic form, such as  $H_3O^+Cl^-$  (Eigen cation) or  $H_5O_2^+Cl^-$  (Zundel cation) formed by spontaneous ionization of adsorbed HCl on ice, up to completion at 150 K.

*Referee Comment:* 4. On page 5 the authors question the importance of the dangling O=H bonds during the early uptake of HCl at low temperature. Is this consistent with the Parent infrared data in their 2011 paper in which they show/remark that the dH-band is gone after 0.15 ML at 50 K. Why did you make the statement WITHOUT GIVING A REASON?

*Author's Response*: We have not emphasized the behavior of the dangling bonds on lines 169-171 being at variance with our unpublished results obtained by Flückiger and Delval (2003). We have inserted a reference to this effect on lines 745-748 to alert to the fact that in our hands the dangling hydrogen bonds are not the locus of adsorption, at least on our ice substrate and deposition conditions at very low HCl partial pressure (T = 120 K). It is possible that the adsorption rate at these low HCl partial pressures is larger for surface sites other than the dangling hydrogen bond sites (dH).

*Author's Changes to manuscript*: Furthermore, the results indicate that the "dangling bonds" of the ice surface attributed to isolated OH groups are not the unique site of HCl adsorption, even in the range 20-90 K (Flückiger and Delval, 2002).

*Insert into bibliographic section* (lines 751-755): Flückiger, B.; Delval, C. Unpublished observations on the behavior of dangling hydrogen bonds (dH) in the presence of HCl at T < 120 K (2002). In essence, the dH absorption intensity at 3396 cm-1 did not decrease in the presence of small HCl partial pressures on the order of 3 x 10-6 Torr at ambient temperature or 4 ppb (Fig. 2.16, Christophe Delval, PhD Thesis no. 3159, EPFL (2005)).

**Referee 1 (Anonymous)**

The answer to many of this referee's technical questions may be found in the bibliographic reference Delval and Rossi (2005) of the revised manuscript. We do not want to take up the publication space of Atmospheric Chemistry and Physics to just copy parts of this manuscript in answering the questions of referee 1. In addition, most of the points raised by referee 1 have already been mentioned in the original text. However, the referee questions/comments show us where we could have been clearer in our explanations, and we nevertheless thank the referee for his diligence and attention to detail. We have emphasized or completed our original text in many locations. Here, we will give brief answers to the specific questions raised by the referee.

*Referee Comment:* Limited Discussion: The manuscript tends to stop at the level of reporting the results without relating them to the results by other groups or lifting them to a more general level.

*Author's Response*: There are simply no other measurements of absolute desorption rates of molecules constituting the components of contaminated ices of atmospheric relevance. All existing reports on desorption rates  $J_{des}(M)$  of H2O, HCl, HNO3, adsorbed organics and the like make the assumption of unity accommodation coefficients which may be wrong by up to three orders of magnitude, whereas the present results were obtained without the incidence of readsorption owing to the small absolute pressures used. The only other study with which to compare the results of HCl is our own dealing with HNO3 absolute rate of desorption referenced in the bibliography (Delval and Rossi, 2005). We will therefore emphasize this point in the discussion section.

Author's Changes to manuscript: None, except that we inserted many explanatory details throughout the text.

*Referee Comment:* No Relevance Given: The introduction is very interesting to read and reveals a detailed discussion on key-topics relevant to the ice-HCl system. However, questions key to this study are not covered: + Why do we need to know J(des)?

*Author's Response*: The value of  $J_{ev}(H_2O)$  ( $J_{des}(H_2O)$ ) is directly related to the net evaporative lifetime once the relative humidity (rh in %) is known, or in the present case, to lifetime prolongation of contaminated atmospheric ice particles with respect to pure water ice. The reciprocal value  $1/J_{ev}(H_2O)(100\text{-rh})$  scales with the time it takes to evaporate a given mass of water ice at a given temperature and rh value and is much longer than for pure ice of equal mass as pointed out in the "Atmospheric Implications Section" (5.). As noted above, there are no measured lifetimes of ice particles except those based on vapor pressures in conjunction with a unity accommodation coefficient that lead to significantly shortened lifetimes of ice particles. Pure ice particles evaporate much too fast so as to disable any significant heterogeneous chemical processes at their interface.

*Author's Changes to manuscript*: None. We recall the sentence spelled out in Section 5. Atmospheric Implications on lines 599-603 including given references: In conclusion we may state that owing to the lifetime extension of ice particles contaminated by HCl, HNO3 or other volatile atmospheric trace gases such as HOCl, HOBr or HONO small particles may have a chance to survive

subsaturated regions of the atmosphere so as to function as cloud condensation or ice nuclei for the following cloud cycle (Delval and Rossi, 2004; 2005; Pratte et al., 2006).

*Referee Comment:* + Where and when is the lifetime of ice particles critical and is the water flux the determining factor

Author's Response: Lifetimes are important when gauging heterogeneous chemistry on ice particles with reactive "reservoir" species in the UT/LS region of the atmosphere as is the case in polar stratospheric chemistry (ozone hole), polar boundary layer ozone disappearance (so-called bromine explosions) or global heterogeneous ozone disappearance on sulfate aerosols in the UT/LS region of the atmosphere. This region of the atmosphere is subject to frequent under- and sometimes oversaturation because the stratosphere is often "dry", that is undersaturated in water vapor.

Author's Changes to manuscript: See previous remark. Regarding the  $H_2O$  saturation level in the UT/LS region and its variation with altitude there is an abundant literature available whose citation is beyond the scope of the present paper. Some aspects are discussed in Delval and Rossi (2005).

*Referee Comment:* + How relevant are the non-equilibrium desorption processes described here to the environment? Please, do not get me wrong. I do believe this lists topics that are nicely addressed by this study and are highly relevant to the environment. It is primalrey the question of discussing those in the text.

*Author's Response*: You can easily convince yourself of the variability of water vapor saturation by looking at contrails that sometimes persist only for a second or so before evaporation (atmosphere heavily undersaturated) or persistent for many hours (atmosphere close to equilibrium w/r to water vapor pressure). What is relatively new is the fact that the UT/LS region is finely structured in that strata (atmospheric layers) of only a few hundred meters thickness change between undersaturation, saturation and supersaturation in water vapor. The balloon-sonde measurements of Terry Deshler some time ago revealed this fine structure in a clear manner. We are refraining from going into meteorology in the present paper and with fundamental physical chemistry of evaporation of H2O and HCl from model atmospheric ice particles which are beyond the scope of the present paper.

Author's Changes to manuscript: None. Please note the bibliographic reference Delva and Rossi 82005) in which some of the questions raised are treated.

*Referee Comment:* Structure: For my feeling, the manuscript jumps to much back and forth between the topics. It is rather difficult to follow. I'd kindly ask you to address these issues and would welcome a revised version. In the following, I give some detailed questions that aim at guiding you. This is not a complete list, and I kindly ask to address the major topics first. A new review can then address the details.

*Author's Response*: Here we take exception to the statement made by referee 1. We cannot and will not rearrange the structure of the paper without more specific advice from this referee. I can assure the referee that much thought has gone into the planning and structuring of the present paper.

Author's Changes to manuscript: None

*Referee Comment:* Introduction, p2: The molecular and dynamic details of crystallization are mentioned. Could you give details on what this would mean for your experiment. What the role of eventually slow formation dynamics in the preparation of your samples, where apparently you start with pure ice to which to dope HCI.

*Author's Response*: The general time scale of the experiment is several to 300 minutes or so. The referee touches upon the thorny question of the molecular mechanism of crystallization and formation of PSC's which out of the present context but is very present in the atmospheric chemistry literature. However, we have not contributed to this problem, let alone performed long-term and low-temperature experiments using both HNO3/H2O, HCl/H2O or any combinations thereof (ternary systems, with and w/o the incidence of H2SO4). We will leave it with the sentence at the end of the first § on page 3 of the Introduction. *Author's Changes to manuscript*: None.

*Referee Comment:* Introduction, p3: Where is the paragraph starting with Fourier-Transform IR heading? What is the take home message with respect to your work?

*Author's Response*: The role of FTIR regarding the present work is part of the Introduction and does not deserve a separate heading. The take-home message may be found under Conclusions starting at line 638. *Author's Changes to manuscript*: None.

*Referee Comment:* Introduction, p3: "Regarding the nature of HCI-ice adsorbate, " What has ionisation to do with your study? This is a long and detailed description in the introduction to which you never return in the discussion.

*Author's Response*: The state of adsorbed HCl is central to the mechanism of heterogeneous reaction including evaporation and condensation of polar gases such as H2O, HCl, HNO3 and H2SO4. *Author's Changes to manuscript*: None.

*Referee Comment:* Experimental: Please specify how did you quantify HCl? How did you derive the mole fraction, i.e. how did you get the volume of ice? Did you assume homogeneous mixture in the total volume of ice? Why is that appropriate? Could you specify on mixing and diffusion times? What is the error on the mole fraction?

*Author's Response*: Please refer to Delval and Rossi (2005) where you may find all details and answers to your technical questions. Regarding a detailed study of diffusion times and mixing of HCl in ice please look at A. Aguzzi, B. Flückiger and M.J. Rossi, Phys. Chem. Chem. Phys. 5, 4157-4169, 2003. Looking at the mass balance data of Table 2 we have inserted new text in lines 277-283:

*Author's Changes to manuscript*: We therefore estimate the average uncertainty  $(2\sigma)$  of the HCl mole fraction  $\chi_{HCl}$  to  $\pm 18\%$  from the average discrepancy between N*dep**HCl* and N*evap**HCl* displayed in Table 2.

Referee Comment: Results, p 7 The average mole fraction should be called "apparent"?

*Author's Response:* It is an apparent quantity, however, we feel that "average" is more suitable because the reader knows what we have done, namely averaging! *Author's Changes to manuscript:* None.

*Referee Comment:* Discussion:Please add discussion of other work on H2O Fluxes from ice in presence of acidic gases. Can your findings be related to water fluxes from other surfaces? Is this result part of a larger picture?

Author's Response: We only have two examples, namely  $HNO_3$  (Delval and Rossi, 2005) and HCl (this work). There are no independent literature results yet. It is hoped that additional measurements will be performed in the near future.

Author's Changes to manuscript: None.

[revised manuscript text omitted]

| 152 | ionic form, such as H 3 O + Cl - (Eigen cation) or H 5 O 2 + Cl - (Zundel cation) formed by |
|-----|--------------------------------------------------------------------------------------------------------------------------------------------------------------------------|
| 153 | spontaneous, ionization of adsorbed HCl on ice, up to completion at 150 K (). The newest                                                                                 |
| 154 | work by Parent compares NEXAFS with photoemission (UPS, XPS) and FTIR in                                                                                                 |
| 155 | transmission of thin HCl/H2O films (Parent et al., 2011). The results are roughly consistent                                                                             |
| 156 | but surprising in the sense that these workers find 92% ionically dissolved HCl in/on ice at 50                                                                          |
| 157 | K in contrast to Kang et al. (2000) and Devlin et al. (2000) under similar exposure (dose) and                                                                           |
| 158 | temperature conditions. In addition, Parent et al. (2011) perform the NEXAFS experiment on                                                                               |
| 159 | a (thick) 100 ML "crystalline" H 2 O ice substrate deposited at 150 K whereas the                                                                             |
| 160 | photoemission and FTIR absorption experiments used a 4 ML thin ice slab deposited at 120                                                                                 |
| 161 | K. The question has to be raised whether the two types of used ice films may be responsible                                                                              |
| 162 | for some of the discrepancies in the results because both the density and the structure of ice                                                                           |
| 163 | are known to be a strong function of temperature and deposition conditions (Kuhs et al., 2012;                                                                           |
| 164 | Schriver-Mazzuoli et al., 2000). The most recent work of Parent et al. (2011) sparked an                                                                                 |
| 165 | interesting controversy in the assignment of the FTIR absorption spectrum of thin HCl/H2O                                                                                |
| 166 | films and led to two comments showcasing the difficulties of intercomparison of nominally                                                                                |
| 167 | identical experiments (Devlin and Kang, 2012; Parent et al., 2012).                                                                                                      |
| 168 | Furthermore, the results indicate that the "dangling bonds" of the ice surface attributed                                                                                |
| 169 | to isolated OH groups are not the unique site of HCl adsorption, even in the range 20-90 K                                                                               |
| 170 | (Flückiger and Delval, 2002). The present, work suggests that maiden uptake of HCl onto                                                                                  |
| 171 | pure ice weakens and perturbs the crystal structure of the ice matrix in an irreversible way                                                                             |
| 172 | such that additional sites for HCl adsorption and ionization are created akin to Parent et al.                                                                           |
| 173 | (2011). Initial HCl uptake on pure ice therefore has a catalytic effect on the following HCl                                                                             |
| 174 | uptake. This irreversible nature of initial HCl dosing is known for several years and has been                                                                           |
| 175 | observed, some time ago in Knudsen flow reactor studies on the HCl/H2O system under                                                                                      |
| 176 | steady-state conditions of both HCl and H2O at temperatures representative of the UT/LS                                                                                  |
| 177 | (Flückiger et al., 1998; Oppliger et al., 1997). The most recent experimental work on                                                                                    |
| 178 | HCl/H2O at an atmospherically relevant ("warm") temperature (253 K) has examined the HCl                                                                                 |
| 179 | depth profile using XPS spectroscopy and finds molecularly adsorbed (physisorbed) HCl at its                                                                             |
| 180 | outermost layer and ionic dissociation in deeper layers (Kong et al., 2017). Complementary                                                                               |
| 181 | X-Ray absorption results also point towards a perturbation of the crystal structure of ice in the                                                                        |
| 182 | aftermath of HCl adsorption/dissolution into deeper layers of ice.                                                                                                       |
| 183 | We have concluded from recent work that HCl doping in quantities of submonolayer                                                                                         |
| 184 | to several monolayers of HCl leads to the decrease of both the evaporative flux Jev (molecule                                                                            |

| 101 | to several monoragers of fiel leads to the decrease of both the evaporative has sev                                                                                                     |  |
|-----|-----------------------------------------------------------------------------------------------------------------------------------------------------------------------------------------|--|
| 185 | cm -2 s -1 ) or rate $R_{ev}$ (molecule cm -3 s -1 ) and the rate of condensation $k_{cond}$ (s -1 ), of H 2 O in the |  |

[revised manuscript text omitted]

ish (United States) ish (United States) Bold

| 855 | Tolbert, M. A.; Rossi, M. J.; Malhotra, R.; Golden, D. M. Reaction of Chlorine Nitrate with |
|-----|---------------------------------------------------------------------------------------------|
| 856 | Hydrogen Chloride and Water at Antarctic Stratospheric Temperatures, Science 1987,          |
| 857 | 238, 1258-1260.                                                                             |

- Uras, N.; Rahman, M.; Devlin, J. P. Covalent HCl at the Surface of Crystalline Ice at 125 K:
  The Stable Phase at Submonolayer Levels J. Phys. Chem. B 1998, *102*, 9375-9377.
- WMO (World Meteorological Organization) *Scientific Assessment of Ozone Depletion 2002*,
  Global Ozone Research and Monitoring Project, report no. 47, Geneva, 2003.
- Xueref, I.; Dominé, F. FTIR spectroscopic studies of the simultaneous condensation of HCl
  and H2O at 190K Atmospheric applications Atmos. Chem. Phys. 2003, *3*, 17791789.
- Yoon, Y. K.; Carpenter, G. B. The Crystal Structure of Hydrogen Chloride Monohydrate,
  Acta Cryst. 1959, *12*, 17-20.
- Zerefos, C. S.; Eleftheratos, K.; Balis, D. S.; Zanis, P.; Tselioudis, G.; Meleti, C. Evidence of
  impact of aviation on Cirrus cloud formation, Atmos. Chem. Phys. 2003, *3*, 16331644.
- 870
- 871
- 872 The authors declare no competing interests regarding the present work.
- 873
- 874 Acknowledgement

875 We sincerely thank Dr. R. Iannarelli for Figures A3 and A4 displayed in the Appendix. We

also would like to thank the Swiss National Science Foundation (SNSF) for unfailing support

- over the years. This work has been performed under SNSF grants no. 20-65299.01 and
- 878 200020-105471.

| Table 1: Hardware parameters of both cryogenic sample supports                                                                                       | QCM                                                                                                                                                                                       |                                                                                                                         |                      |  |             |  |
|------------------------------------------------------------------------------------------------------------------------------------------------------|-------------------------------------------------------------------------------------------------------------------------------------------------------------------------------------------|-------------------------------------------------------------------------------------------------------------------------|----------------------|--|-------------|--|
| Reactor temperature T r [K]                                                                                                               | 32                                                                                                                                                                                        | 20                                                                                                                      |                      |  |             |  |
| Reactor volume V r [cm -3 ]                                                                                                    | 50                                                                                                                                                                                        |                                                                                                                         |                      |  |             |  |
| Conversion factor (1/RT) Conv [molec cm -3 Torr -1 ]
with R=62398 [Torr cm 3 mol -1 K -1 ] | 0 16 (1)                                                                                                                                                                       |                                                                                                                         |                      |  |             |  |
| Sample surface area [cm 2 ]                                                                                                               | 0.78                                                                                                                                                                                      | 0.50                                                                                                                    |                      |  |             |  |
| $H_2O$ collision frequency with ice sample $\omega_{H_2O}$ [s -1 ]                                                                        | 5.08                                                                                                                                                                                      | 3.26                                                                                                                    |                      |  |             |  |
| H 2 O effusion rate constant of calibrated leak k esc (H 2 O) [s -1 ]                                    | 0.0                                                                                                                                                                                       | 64                                                                                                                      |                      |  |             |  |
| MS calibration factor for H 2 $\underline{O}_{(m/z=18, Stirred Flow)} C_{18}^{S-Flow}$ [molec s -1 A -1 ]           | 2.4 •                                                                                                                                                                                     | 10 24                                                                                                        |                      |  | Deleted: 0  |  |
| MS calibration factor for H 2 $\underline{O}$ (m/z=18, Dynamic ) $C_{18}^{dyn}$ [molec s -1 A -1 ]                  | 1.7 •                                                                                                                                                                                     | 10 25                                                                                                        |                      |  | Deleted: 0  |  |
| HCl collision frequency with ice sample $\omega_{HCl}$ [s -1 ]                                                                            | 3.59                                                                                                                                                                                      | 2.31                                                                                                                    |                      |  |             |  |
| HCl effusion rate constant of calibrated leak kesc(HCl) [s -1 ]                                                                           | 0.0                                                                                                                                                                                       | 0.047                                                                                                                   |                      |  |             |  |
| MS calibration factor for HC1 (m/z=36, Stirred Flow) $C_{36}^{S-Flow}$ [molec s -1 A -1 ]                                      | 10 24                                                                                                                                                                          |                                                                                                                         |                      |  |             |  |
| MS calibration factor for HC1 (m/z=36, Dynamic) C dyn 36 [molec s -1 A -1 ]                              | 6.3 •                                                                                                                                                                                     | 10 24                                                                                                        |                      |  |             |  |
| Calculated escape orifice area A esc [mm 2 ]                                                                                   | 1.                                                                                                                                                                                        | 0                                                                                                                       |                      |  |             |  |
|                                                                                                                                                      | $\label{eq:d} \begin{split} d &= 10^4  \text{\AA} \\ & \text{or } 1.0 \; \mu \text{m for} \\ & \text{O.D.} = 1.08 \; [\text{A}]^{(2)} \\ & \text{at } 3260 \; \text{cm}^{-1} \end{split}$ | $d = 10^{4} \text{ Å}$
or 1.0 µm for
O.D.= 1.08 [A] (2)
at 3260 cm -1 Calibration Factor |                      |  |             |  |
|                                                                                                                                                      |                                                                                                                                                                                           | Temperature [K]                                                                                                         | ratio (3) |  |             |  |
| 1 Wall temperature of the reactor at $T = 320$ K                                                                                          |                                                                                                                                                                                           | 170                                                                                                                     | 9.0                  |  |             |  |
| 2 See, ( Delval et al., 2003 ).                                                                                                    |                                                                                                                                                                                           | 180                                                                                                                     | 8.0                  |  | Deleted: R  |  |
| 3 Corresponds to the ratio between the true number of molecules present on the QCM                                                        |                                                                                                                                                                                           | 190                                                                                                                     | 7.8                  |  | Deleted: [  |  |
| support and the number of molecules displayed by the IC5 controller (Delval et al.,                                                           |                                                                                                                                                                                           | 193                                                                                                                     | 6.0                  |  | Deleted: 41 |  |
| 2004)                                                                                                                                                |                                                                                                                                                                                           | 205                                                                                                                     | 2.0                  |  | Deleted: ]  |  |
| ۲                                                                                                                                                    |                                                                                                                                                                                           | 208                                                                                                                     | 1.9                  |  | Deleted:    |  |

eference Deleted: [ Deleted: 42

31

| Experiment | T ice | do    | $N^0_{H2O}$         | R HCl         | t dep | N dep HCl | HCl  | N evap
HCl | $\chi^{0}_{HCl}$       | dD    | $J^{b}_{ev}$                              | $J^{e}_{ev}$                              | r b/e |
|------------|------------------|-------|---------------------|--------------------------|------------------|---------------------------------|------|--------------------------|------------------------|-------|-------------------------------------------|-------------------------------------------|------------------|
| number     | [K]              | [Å]   | [molec]             | [molec s -1 ] | [s]              | [molec]                         | ML   | [molec]                  |                        | [Å]   | [molec cm -2 s -1 ] | [molec cm -2 s -1 ] |                  |
| 10         | 174              | 15230 | $2.4 \cdot 10^{18}$ | $6.4 \cdot 10^{12}$      | 94               | 6.0 · 10 14          | 4.8  | $4.7 \cdot 10^{14}$      | $2.5 \cdot 10^{-4}$    | 2733  | $1.9 \cdot 10^{15}$                       | $4.4 \cdot 10^{14}$                       | 4.3              |
| 5          | 188              | 13318 | $2.0 \cdot 10^{18}$ | $1.3 \cdot 10^{13}$      | 66               | $8.7 \cdot 10^{14}$             | 7.0  | $8.9 \cdot 10^{14}$      | $4.4 \cdot 10^{-4}$    | 4540  | $1.2 \cdot 10^{16}$                       | $3.9 \cdot 10^{15}$                       | 3.1              |
| 4          | 190              | 14016 | $2.1 \cdot 10^{18}$ | $4.2 \cdot 10^{13}$      | 126              | $5.4 \cdot 10^{15}$             | 43.2 | 3.6 · 10 15   | $2.6 \cdot 10^{-3}$    | 6360  | 2.9 · 10 16                    | $1.4 \cdot 10^{15}$                       | 20.7             |
| 6          | 190              | 13886 | $2.1 \cdot 10^{18}$ | $3.9 \cdot 10^{13}$      | 56               | $2.2 \cdot 10^{15}$             | 17.6 | $1.8 \cdot 10^{15}$      | $1.0 \cdot 10^{-3}$    | 12861 | $3.4 \cdot 10^{16}$                       | $1.7 \cdot 10^{16}$                       | 2.0              |
| 1          | 192              | 14926 | $2.3 \cdot 10^{18}$ | $3.1 \cdot 10^{12}$      | 36               | $1.0 \cdot 10^{14}$             | 0.8  | $1.8 \cdot 10^{14}$      | $4.3 \cdot 10^{-5}$    | 2823  | $2.9 \cdot 10^{16}$                       | $7.1 \cdot 10^{14}$                       | 40.8             |
| 2          | 192              | 14682 | $2.3 \cdot 10^{18}$ | 8.0 · 10 11   | 356              | 2.6 · 10 14          | 2.1  | $1.6 \cdot 10^{14}$      | 1.1 • 10 -4 | 6817  | $3.2 \cdot 10^{16}$                       | $6.5 \cdot 10^{14}$                       | 49.2             |
| 11         | 192              | 14420 | $2.2 \cdot 10^{18}$ | $5.4 \cdot 10^{12}$      | 108              | 5.4 · 10 14          | 4.3  | 6.8 · 10 14   | $2.4 \cdot 10^{-4}$    | 7717  | $4.0 \cdot 10^{16}$                       | $7.9 \cdot 10^{14}$                       | 50.6             |
| 3          | 193              | 14423 | $2.2 \cdot 10^{18}$ | $3.5 \cdot 10^{12}$      | 220              | $7.0 \cdot 10^{14}$             | 5.6  | 8.1 · 10 14   | $3.2 \cdot 10^{-4}$    | 5659  | $4.9 \cdot 10^{16}$                       | $1.8 \cdot 10^{15}$                       | 27.2             |
| 7          | 195              | 12614 | $1.9 \cdot 10^{18}$ | $4.3 \cdot 10^{12}$      | 45               | $1.9 \cdot 10^{14}$             | 1.5  | $1.8 \cdot 10^{14}$      | $1.0 \cdot 10^{-4}$    | 5325  | $4.6 \cdot 10^{16}$                       | 2.0 · 10 15                    | 23.0             |
| 8          | 205              | 13505 | $2.1 \cdot 10^{18}$ | $1.6 \cdot 10^{13}$      | 36               | 5.9 · 10 14          | 4.7  | 3.0 · 10 14   | $2.8 \cdot 10^{-4}$    | 4607  | 2.0 · 10 17                    | $1.0 \cdot 10^{16}$                       | 20.0             |
| 9          | 210              | 13134 | $2.0 \cdot 10^{18}$ | 3.5 · 10 12   | 84               | 3.0 · 10 14          | 2.4  | $1.9 \cdot 10^{14}$      | 1.5 · 10 -4 | 12136 | 3.0 · 10 17                    | $1.8 \cdot 10^{16}$                       | 16.7             |

 Table 2 : Representative experimental results for the kinetics of H2O evaporation in the presence of HCl for increasing HCl deposition temperatures at given rates of deposition RHCl and doses of HCl NHCldep. In the first column the number refers to the corresponding experiment and identifies the data displayed in Figure 2

**Figure Captions**

Figure 1: Typical experimental protocol of the evaporation at 192 K of an approximately 1.2  $\mu$ m thick ice film doped with 5.4 $\cdot$ 1014 molecules of HCl. This illustration corresponds to experiment 11 of Table 2. ( $\circ$ ): ice thickness monitored by QCM (Å), ( $\Box$ ): "apparent" H2O evaporative flux, JQCMev, monitored by QCM (molecule cm-2 s-1), (+): I18 MS signal for H2O, (×): I36 MS signal for HCl (A), ( $\diamond$ ): J18ev evaporative flux calculated from I18 (molecule cm-2 s-1), ( $\Delta$ ): Int(J18ev) time integral of J18ev (molecule cm-2).

Figure 2: Change of the evaporative flux  $J_{ev}(H_2O)$  as a function of the HCl mole fraction ( $\chi_{HCl}$ ) for the cases presented in Table 2 color-coded according to the corresponding experiment number in Table 2. The colored and circled numbers on axis "b" (left) correspond to  $J_{ev}(H_2O)$  of pure ice before HCl deposition, the ones on axis "e" (right) are  $J_{ev}(H_2O)$  at t = tHe at the end of HCl evaporation. The colored circles in the data field mark the value of  $J_{ev}(H_2O)$  after HCl deposition at t = t0 and are equal to  $J_{ev}(H_2O)$  of pure ice. The start of any particular  $J_{ev}(H_2O)$  curve as a continuous solid (bold) line occurs at t = tD at 85% of  $J_{ev}(H_2O)$  at t = 0 (pure ice value, colored circle or circled number on axis "b" to the left) and ends at tHb, the beginning of HCl evaporation as displayed in Figure 1B.

Figure 3: Synopsis of the dependence of the evaporation range parameter  $r^{b/e}$  on the rate of deposition  $R_{HCl}$  of HCl for temperatures between 188 and 210 K. Each point is marked with the total number of HCl molecules ( $N_{HCl}$ ) deposited on the ice film , the temperature of the ice film at HCl deposition and the experiment number (bold) referring to Table 2. The hashed area encompasses  $r^{b/e}$  values for dataset B (experiments 3,4,7,8). The color code goes from low (blue) over medium (green) to high (red) temperatures.

Figure 4: Synopsis of the dependence of  $d_D$  on the rate of deposition  $R_{HCl}$  of HCl for temperatures between 188 and 210 K. Each point is marked with the total number of HCl molecules ( $N_{HCl}$ ) deposited on the ice film, the temperature of the ice film at HCl deposition and the experiment number (bold font) referring to Table 2. The hashed area encompasses  $d_D$ values for dataset B (experiments 3,4,7,8). The color code goes from low (blue) over medium (green) to high (red) temperatures.

---

## Author Response (AR2)

**Point-by-Point Reply to Co-Editor of acp manuscript 2018-238 (author) –**
**minor revisions**
We thank the anonymous referee for his diligence regarding the second round of reviews of the above-
referenced manuscript. We have carefully reread the entire manuscript, made corrections/additions
here and there (highlighted in the track changes copy) and hope that we may satisfy the publication
requirements as proposed in the following:
*Reviewer: I still feel that the introduction does not prepare the reader well enough for the results and discussion.*
*On the one hand, the topic of acid dissociation is which covers more than a page of the manuscript is not at all*
*taken up in the discussion. Even worse, taken that the atmospheric implications (5) discuss the lifetime of cloud*
*particles this seems to be the central science tackled by the research. The introduction does not even once*
*mention lifetime (except when giving goals of this paper) – I think you need to explain to the reader why this is*
*important and which aspects are open concerning lifetime of ice particles in the atmosphere. In summary, the*
*open questions around the water flux are still not clearly presented in the introduction, or lost in the details of*
*the text, and the new aspects of this work relative to the previous publications could be more emphasized.*
*However, I leave this decision to the editor.*
Author answer: We have tried to focus on two fundamental criteria regarding the occurrence of
heterogeneous chemical reactions on atmospheric ices, namely (a) absolute evaporation rates of H2O
in water-rich ices, and (b) elementary reaction kinetics of the interfacial processes yielding the
precursors to atomic halogens.
Addition to text: Regarding heterogeneous chlorine activation leading to polar ozone loss in the LS
(Solomon et al., 1986), but also tropical ozone depletion in the UT/LS region, there are two
fundamental questions that continue to be raised, namely, (a) Do the involved heterogeneous
reactions have sufficient time to proceed on the surface (interfacial chemical mechanism) or in the
volume (bulk mechanism) of atmospheric ice particles? and (b) Are the heterogeneous reactions
sufficiently rapid for the formation of the precursors of atomic chlorine under LS or UT/LS atmospheric
conditions? Question (a) is directly related to the evaporative lifetime or the absolute rate of
evaporation of $H_2O$, the main component of atmospheric ices, whereas (b) mainly depends on the
chemical mechanism of the heterogeneous chlorine activation reaction and the ionic vs. molecular
state of adsorbed HCl, either in the form of ionic hydrates ($H_3O^+(H_2O)_nCl^-$) or molecularly adsorbed
HCl. Ionic reactions in a polar medium do not significantly slow down with lowering the temperature
whereas reactions between closed shell molecules such as HCl and $ClONO_2$ or other molecules often
have small barriers and thus are activated which leads to a significant decrease of the reaction rate
with decreasing temperature. The present work primarily addresses the measurement of absolute
rates of evaporation of $H_2O$ from atmospheric ices that therefore directly impact the lifetimes of atmospheric ice particles under subsaturated atmospheric conditions. This enables the evaluation
whether or not heterogeneous processing will take place to a significant extent taking into account
published reports in the literature on the state of adsorbed HCl which is briefly summarized below.
*Reviewer: L 161 ff: "The question has to be raised whether the two types of used ice films may be responsible*
*some of the discrepancies in the results because both the density and the structure of ice are known to be a*
*strong function of temperature and deposition conditions (Kuhs et al., 2012; Schriver-Mazzuoli et al., 2000).*
*This line gives the discussion of HCl dissociation a good reason with relevance to this work. I wonder, though,*
*why the question of dissociation is not picked up in the discussion again taken that it occupies a rather large*
*part of the introduction.*
Author Reply: By focusing on two fundamental criteria (see previous point) we create a link to the
presentation of the adsorption state of HCl on ice. Question (a) (see previous) is treated in great detail
in this work (evaporative lifetime), whereas question (b) is taken from literature results (adsorption
state of HCl on ice). This is one of the reasons that we have not taken up this topic in the Discussion
because we have not contributed to this subject in the present work although our FTIR spectra are
consistent with the interpretation found in the literature.
No text added.
*Reviewer: L 199 ff: "It appears that two observed HCl species on/in ice, namely single phase amorphous*
*HCl/$H_2O$ mixtures and a binary phase consisting of pure ice and an as yet unidentified crystalline HCl hydrate,*
*HCl•xH2O, decrease $J_{ev}(H_2O)$ to a different extent as proposed in Delval et al. (2003)." Appear to me very close*
*to the conclusions a, b, and e. May I ask you to stress the new findings in slight more detail.*
Author Reply: We are accommodating to the request of the reviewer by inserting the following text:
Text Addition: We believe that the unidentified crystalline HCl hydrate observed in Delval et al.
(2003) corresponds to a metastable form of HCl hexahydrate because the bending vibration of the
proton-ordered waters of hydration are split into a doublet (1644, 1618 cm$^{-1}$) in this work whereas all
others report a single sharp peak in the range 1635 to 1643 cm$^{-1}$ (see Table 2 in Delval et al., 2003).
*Reviewer: L 288: The average mole fraction χHCl of HCl in the remaining ice film as a function of time is*
*calculated according to Delval and Rossi (2005). Please detail on the method. Is this a direct measure of HCl*
*deposited to the sample or rather indirect based on the gas-phase signal (dosage).*

Author Reply: We have entered the below-listed sentences into the text in order to briefly explain how the HCl mole fractions were determined.

Added text: Briefly, the number of HCl molecules adsorbed onto a given thin ice film consisting of a known number of $H_2O$ molecules using the calibrated signal of the QCMB has been established as a difference between the calibrated MS signals at m/e 36 of the quartz crystal sensor at ambient minus the (low) temperature of measurement akin to Figure 3 in Delval et al. 2005. Column 7

of Table 2 ($\mathbf{N}_{HCl}^{dep}$) displays the corresponding values. The HCl mole fraction $\chi^0_{HCl}$ has been established as the ratio $\mathbf{N}_{HCl}^{dep}$ / $\mathbf{N}_{H2O}^{0}$ using the approximation $\mathbf{N}_{HCl}^{dep}$ << $\mathbf{N}_{H2O}^{0}$ which is justified in this case as may be seen from the numbers displayed in column 4 and 7 of Table 2. This amounts to an absolute measurement of the number of HCl molecules lost or adsorbed onto the cold quartz sensor in contrast to a simple dose (exposure) measurement.

*Reviewer: Figure 2. Why are experiments 5 and 6 not discussed. Would this be a third type of ice?*

Author Reply: We clearly talk about limiting behavior of $J_{ev}(H_2O)$ by two defined states of HCl, namely crystalline HCl hydrate and amorphous $HCl/H_2O$ liquid. Experiments 5 and 6 seem to lie between these two limiting cases which makes the interpretation difficult. It may be an external or internal mixture of the crystalline and the amorphous phase, but we cannot say at the moment as we do not control the growth conditions.

*Reviewer: Figure 2. Why is experiment 10 not shown even thought it is discussed in the text? Please add.*

Author Reply: Experiment 10 is presented in Table 2. The reason we have not included it in Figure 2

is that it is not particularly compelling. It just makes the drawing even more crowded such that we decided to omit it from the graph as it does not confer any new information.

**The Influence of HCl on the Evaporation Rates of H$_2$O over Water Ice in**

**the Range 188 to 210 K at small Average Concentrations**

Christophe Delval[1,2,a] and Michel J. Rossi[1,3]

[1]Laboratory of Air and Soil Pollution Studies (LPAS), ENAC Faculty, Swiss Federal Institute of Technology (EPFL), CH-1015 Lausanne, Switzerland

[2]Atmospheric Particle Research Laboratory (APRL), ENAC Faculty, Swiss Federal Institute of Technology (EPFL), CH-1015 Lausanne, Switzerland

[3]Laboratory of Atmospheric Chemistry (LAC), Paul Scherrer Institute (PSI), CH-5232

Villigen-PSI, Switzerland

[a]Present Address: Patent Examiner - Directorate 1657, Dir. 1.6.5.7, European Patent Office,

Patentlaan 3-9, 2288 EE Rijswijk, Netherlands

Corresponding author:  michel.rossi@psi.ch

**ABSTRACT**

  The evaporation flux $J_{ev}(H_2O)$ of $H_2O$ from HCl-doped typically 1.5 μm or so thick vapor- deposited ice films has been measured in a combined quartz crystal microbalance (QCMB) – residual gas mass spectrometry (MS) experiment using a stirred flow reactor (SFR). $J_{ev}(H_2O)$ has been found to show complex behaviour and to be a function of the average mole fraction $\chi_{HCl}$ of HCl in the ice film ranging from $6\times10^{14}$ to $3\times10^{17}$ molecule cm$^{-2}$ s$^{-1}$ at 174 – 210 K for initial values $\chi^0_{HCl}$ ranging from

$5\times10^{-5}$ to $3\times10^{-3}$ at the start of the evaporation. The dose of HCl on ice was in the range of 1 to 40

formal monolayers and the $H_2O$ vapor pressure was independent of $\chi_{HCl}$ within the measured range and equal to that of pure ice down to 80 nm thickness. The dependence of $J_{ev}(H_2O)$ with increasing average $\chi_{HCl}$ was correlated with (a) the evaporation range r$^{b/e}$ parameter, that is the ratio of $J_{ev}(H_2O)$

just before HCl-doping of the pure ice film and $J_{ev}(H_2O)$ once most of HCl has desorbed towards the end of film evaporation, and (b) the remaining thickness $d_D$ below which $J_{ev}(H_2O)$ decreases to less than 85% of pure ice. The dependence of $J_{ev}(H_2O)$ with increasing average $\chi_{HCl}$ from HCl-doped ice films suggests two limiting data sets: One consisting of a two-phase pure ice/crystalline HCl hydrate binary phase (set A), and the other of a single phase amorphous HCl/$H_2O$ binary mixture (set B). The measured values of $J_{ev}(H_2O)$ may lead to significant evaporative life-time extensions of HCl- contaminated ice cloud particles under atmospheric conditions, regardless of whether the structure corresponds to an amorphous or crystalline state of the HCl/$H_2O$ aggregate.

1. **INTRODUCTION**

HCl is among the mineral acids that control the acidity of the atmosphere together with $HNO_3$ and $H_2SO_4$. The production of atmospheric HCl is predominantly taking place in the middle and upper stratosphere where $O_3$ is formed owing to photolysis of halogen containing source gases such as CFC's (chlorofluorocarbons). However, there are no known sources of HCl in the upper troposphere (UT) because scavenging processes of HCl throughout the troposphere are very efficient which leads to HCl background concentrations of less than 0.1 ppb (Graedel and Keene, 1995). The absence of significant sources in the troposphere, the long photolytic lifetime of HCl and the fact that the production region is well separated from the regions of interest, namely the UT and the lower stratosphere (LS) all contribute to the fact that HCl is an excellent tracer for stratospheric ozone in the UT (Marcy et al., 2004). Owing to the frequent occurrence of Cirrus clouds in this atmospheric region it is of obvious interest to study the interaction of HCl with atmospheric ice particles at relevant temperature and pressure conditions (Jensen et al. 2001; Zerefos et al., 2003). The compact correlation between $O_3$ and HCl has been used to monitor stratospheric-tropospheric exchange processes and stratospheric $O_3$ intrusions into the troposphere that are still an active field of investigation (Houghton et al., 2001).

HCl is of importance in the LS as it partakes in heterogeneous reactions on Polar

Stratospheric Clouds (PSC's consisting either of nitric acid trihydrate, $HNO_3 \cdot 3H_2O$ or Cirrus clouds), on Cirrus clouds in the UT/LS region (Borrmann et al., 1996) as well as on background stratospheric $H_2SO_4$ aerosol according to the following reaction taken as an example of the many possible heterogeneous processes:

$HCl(ads) + ClONO_2 \rightarrow Cl_2(g) + HNO_3(ads)$

These reactions efficiently convert inactive Cl-containing reservoir molecules such as HCl and $ClONO_2$ into active photolyzable Cl-containing compounds in a single reaction. Typical examples of such photolabile reaction products are $Cl_2$, $ClNO_2$ and HOCl that will change the atmospheric composition owing to the high reactivity of the photolysis products such as atomic Cl (Solomon et al., 1986; Tolbert et al., 1987; WMO 2003). It thus follows that HCl is of stratospheric (LS) importance and is frequently used as a model compound for heterogeneous chlorine activation reactions on ices that has inspired many laboratory kinetic studies (Leu et al., 1991; Hanson and Ravishankara, 1992; Chu et al., 1993; Flückiger et al.,

1998; Hynes et al., 2001; Abbatt, 2003).

Regarding heterogeneous chlorine activation leading to polar ozone loss in the LS
(Solomon et al., 1986), but also ozone depletion in the tropical UT/LS region, there are two
fundamental questions that continue to be raised, namely, (a) Do the involved heterogeneous
reactions have sufficient time to proceed on the surface (interfacial chemical mechanism) or
in the volume (bulk mechanism) of atmospheric ice particles? and (b) Are the heterogeneous
reactions sufficiently rapid for the formation of the precursors of atomic chlorine under LS or
UT/LS atmospheric conditions? Question (a) is directly related to the evaporative lifetime or
the absolute rate of evaporation of $H_2O$, the main component of atmospheric ices, whereas (b)
mainly depends on the chemical mechanism of the heterogeneous chlorine activation reaction
and the ionic vs. molecular state of adsorbed HCl, either in the form of ionic hydrates
$(H_3O^+(H_2O)_nCl^-)$ or molecularly adsorbed HCl. Ionic reactions in a polar medium do not
significantly slow down with lowering the temperature whereas reactions between closed
shell molecules such as HCl and $ClONO_2$ or other molecules often have small barriers and
thus are activated which leads to a significant decrease of the reaction rate with decreasing
temperature. The present work primarily addresses the measurement of absolute rates of
evaporation of $H_2O$ from atmospheric ices that therefore directly impact the lifetimes of
atmospheric ice particles under subsaturated atmospheric conditions. This enables the
evaluation whether or not heterogeneous processing will take place to a significant extent
taking into account published reports in the literature on the state of adsorbed HCl which is
briefly summarized below.

[revised manuscript text omitted]
 different extents as proposed in Delval et al. (2003). We believe that the unidentified crystalline HCl hydrate observed in Delval et al. (2003) corresponds to a metastable form of HCl hexahydrate because the bending vibration of the proton-ordered waters of hydration are split into a doublet (1644, cm$^{-1}$) in this work whereas all others report a single sharp peak in the range 1635 to 1643 cm$^{-1}$ (see Table 2 in Delval et al., 2003). These results have led us to perform systematic experiments in this work using the Quartz Crystal MicroBalance (QCMB) combined with residual gas Mass Spectrometry (MS) that we have used successfully in the past (Delval and Rossi, 2004) in order to investigate the temporal change of $J_{ev}(H_2O)$ with increasing average mole fraction of HCl, $\chi_{HCl}$, remaining in the ice. One of the goals of the present work is to determine the influence of the HCl deposition parameters on the temporal change of $J_{ev}$ and the mass accommodation coefficient $\alpha$ during evaporation of a HCl-doped ice film and its consequence on the lifetime of atmospheric ice particles contaminated by HCl. This issue is key in relation to the importance of heterogeneous vs. homogeneous atmospheric reactions at midlatitudes as has been pointed out in the past (Solomon et al., 1986; 1997).

**2. EXPERIMENTAL**

[revised manuscript text omitted]

**3. RESULTS**

        The experimental data reported in Table 2 on the isothermal change of the evaporative flux of water, $J_{ev}(H_2O)$, as a function of the average mole fraction of HCl, $\chi_{HCl}$, in the remaining ice film during the evaporation process under dynamic conditions are presented in Figure 2. Dynamic pumping conditions ensure the absence of any readsorption of $H_2O$ vapor during evaporation owing to the low $H_2O$ partial pressures in the reactor. The axes labelled "b" and "e" correspond to the values of $J_{ev}(H_2O)$ at the end of ice film deposition and after desorption of most of the adsorbed HCl from the HCl-doped ice film at $t_{He}$, respectively, as displayed in Figure 1B. The average mole fraction $\chi_{HCl}$ of HCl in the remaining ice film as a function of time is calculated according to Delval and Rossi (2005). Briefly, the number of HCl molecules adsorbed onto a given thin ice film consisting of a known number of $H_2O$ molecules using the calibrated signal of the QCMB has been established as a difference between the calibrated MS signals at m/e 36 of the quartz crystal sensor at ambient minus the (low) temperature of measurement akin to Figure 3 in Delval et al. 2005. Column 7 of Table 2 ($N^{dep}_{HCl}$) displays the corresponding values. The HCl mole fraction $\chi^0_{HCl}$ has been established as the ratio $N^{dep}_{HCl} / N^0_{H2O}$ using the approximation $N^{dep}_{HCl} << N^0_{H2O}$ which is justified in this case as may be seen from the numbers displayed in column 4 and 7 of Table 2. This amounts to an absolute measurement of the number of HCl molecules lost or adsorbed onto the cold quartz sensor in contrast to a simple dose (exposure) measurement where one does not know the amount of adsorbed molecules.

[revised manuscript text omitted]

HCl on single crystal (SC), condensed $H_2O$ vapour from the gas phase called C-ice and frozen liquid water ice (B for "bulk") at low temperatures. Taking the averaged value for B- and C- ice of $D_{HCl} = 2.3 \times 10^{-13}$ cm$^2$ s$^{-1}$ at 190 K we calculate the average distance taken for 3-D

molecular diffusion of HCl in ice as 83 and 203 nm for a time horizon of 100 and 600 s, respectively (see time scale of Figure 1). This evaluation illustrates the point that during the time scale of the present experiments molecular diffusion of HCl encompasses the typical water ice interface that corresponds to a structurally perturbed region of the macroscopic thin ice films. Of course, HCl will diffuse to deeper layers into the bulk with increasing temperature taking into account the exponential dependence of D on temperature (Aguzzi et al., 2003). 
[revised manuscript text omitted]

Aguzzi, A.; Flückiger, B.; Rossi, M.J. The nature of the interface and the diffusion coefficient of HCl/ice and HBr/ice in the temperature range 190–205 K, Physical Chemistry

Chemical Physics **2003**, *5*, 4157-4169.

Banham, S. F.; Horn, A. B.; Koch, T. G.; Sodeau, J. R. Ionisation and solvation of stratospherically relevant molecules on ice films, Faraday Discuss. **1995**, *100*, 321-

332.

Biermann, U.; Crowley, J. N.; Huthwelker, T.; Moortgat, G. K.; Crutzen, P. J.; Peter, T. FTIR

studies on lifetime prolongation of stratospheric ice particles due to NAT coating,

Geophys. Res. Lett. **1998**, *25*, 3939-3942.

Bolton, K.; Petterson, J. B. C. Ice-Catalyzed Ionization of Hydrochloric Acid, J. Amer. Chem.

Soc. **2001**, *123*, 7360-7363.

Borrmann, S.; Solomon, S.; Dye, J.E.; Luo, B. The potential of cirrus clouds for heterogeneous chlorine activation, Geophys. Res. Lett. **1996**, *23*, 2133-2136.

[revised manuscript text omitted]
}$ [K] | $d_0$ [Å] | $N^0_{H2O}$ [molec] | $R_{HCl}$ [molec s⁻¹] | $t_{dep}$ [s] | $N^{dep}_{HCl}$ [molec] | HCl ML | $N^{evap}_{HCl}$ [molec] | $\chi^0_{HCl}$ | $d_D$ [Å] | $J^b_{ev}$ [molec cm⁻² s⁻¹] | $J^e_{ev}$ [molec cm⁻² s⁻¹] | $r^{b/e}$ |
|---|---|---|---|---|---|---|---|---|---|---|---|---|---|
| 10 | 174 | 15230 | $2.4\cdot10^{18}$ | $6.4\cdot10^{12}$ | 94 | $6.0\cdot10^{14}$ | 4.8 | $4.7\cdot10^{14}$ | $2.5\cdot10^{-4}$ | 2733 | $1.9\cdot10^{15}$ | $4.4\cdot10^{14}$ | 4.3 |
| 5 | 188 | 13318 | $2.0\cdot10^{18}$ | $1.3\cdot10^{13}$ | 66 | $8.7\cdot10^{14}$ | 7.0 | $8.9\cdot10^{14}$ | $4.4\cdot10^{-4}$ | 4540 | $1.2\cdot10^{16}$ | $3.9\cdot10^{15}$ | 3.1 |
| 4 | 190 | 14016 | $2.1\cdot10^{18}$ | $4.2\cdot10^{13}$ | 126 | $5.4\cdot10^{15}$ | 43.2 | $3.6\cdot10^{15}$ | $2.6\cdot10^{-3}$ | 6360 | $2.9\cdot10^{16}$ | $1.4\cdot10^{15}$ | 20.7 |
| 6 | 190 | 13886 | $2.1\cdot10^{18}$ | $3.9\cdot10^{13}$ | 56 | $2.2\cdot10^{15}$ | 17.6 | $1.8\cdot10^{15}$ | $1.0\cdot10^{-3}$ | 12861 | $3.4\cdot10^{16}$ | $1.7\cdot10^{16}$ | 2.0 |
| 1 | 192 | 14926 | $2.3\cdot10^{18}$ | $3.1\cdot10^{12}$ | 36 | $1.0\cdot10^{14}$ | 0.8 | $1.8\cdot10^{14}$ | $4.3\cdot10^{-5}$ | 2823 | $2.9\cdot10^{16}$ | $7.1\cdot10^{14}$ | 40.8 |
| 2 | 192 | 14682 | $2.3\cdot10^{18}$ | $8.0\cdot10^{11}$ | 356 | $2.6\cdot10^{14}$ | 2.1 | $1.6\cdot10^{14}$ | $1.1\cdot10^{-4}$ | 6817 | $3.2\cdot10^{16}$ | $6.5\cdot10^{14}$ | 49.2 |
| 11 | 192 | 14420 | $2.2\cdot10^{18}$ | $5.4\cdot10^{12}$ | 108 | $5.4\cdot10^{14}$ | 4.3 | $6.8\cdot10^{14}$ | $2.4\cdot10^{-4}$ | 7717 | $4.0\cdot10^{16}$ | $7.9\cdot10^{14}$ | 50.6 |
| 3 | 193 | 14423 | $2.2\cdot10^{18}$ | $3.5\cdot10^{12}$ | 220 | $7.0\cdot10^{14}$ | 5.6 | $8.1\cdot10^{14}$ | $3.2\cdot10^{-4}$ | 5659 | $4.9\cdot10^{16}$ | $1.8\cdot10^{15}$ | 27.2 |
| 7 | 195 | 12614 | $1.9\cdot10^{18}$ | $4.3\cdot10^{12}$ | 45 | $1.9\cdot10^{14}$ | 1.5 | $1.8\cdot10^{14}$ | $1.0\cdot10^{-4}$ | 5325 | $4.6\cdot10^{16}$ | $2.0\cdot10^{15}$ | 23.0 |
| 8 | 205 | 13505 | $2.1\cdot10^{18}$ | $1.6\cdot10^{13}$ | 36 | $5.9\cdot10^{14}$ | 4.7 | $3.0\cdot10^{14}$ | $2.8\cdot10^{-4}$ | 4607 | $2.0\cdot10^{17}$ | $1.0\cdot10^{16}$ | 20.0 |
| 9 | 210 | 13134 | $2.0\cdot10^{18}$ | $3.5\cdot10^{12}$ | 84 | $3.0\cdot10^{14}$ | 2.4 | $1.9\cdot10^{14}$ | $1.5\cdot10^{-4}$ | 12136 | $3.0\cdot10^{17}$ | $1.8\cdot10^{16}$ | 16.7 |

**Figure Captions**

[revised manuscript text omitted]